# Linear Log-Normal Attention with Unbiased Concentration

**Yury Nahshan, Joseph Kampeas and Emir Haleva**
Distributed and Parallel Software Lab, Huawei Technologies
Email: {first.last}@huawei.com

## Abstract

Transformer models have achieved remarkable results in a wide range of applications. However, their scalability is hampered by the quadratic time and memory complexity of the self-attention mechanism concerning the sequence length. This limitation poses a substantial obstacle when dealing with long documents or high-resolution images. In this work, we study the self-attention mechanism by analyzing the distribution of the attention matrix and its concentration ability. Furthermore, we propose instruments to measure these quantities and introduce a novel self-attention mechanism, Linear Log-Normal Attention, designed to emulate the distribution and concentration behavior of the original self-attention. Our experimental results on popular natural language benchmarks reveal that our proposed Linear Log-Normal Attention outperforms other linearized attention alternatives, offering a promising avenue for enhancing the scalability of transformer models.

## 1 Introduction

Transformer models, proposed by (Vaswani et al., 2017), have become widely used deep learning architectures that have achieved state-of-the-art performance in various fields, including Natural Language Processing (NLP) (Brown et al., 2020; Devlin et al., 2018), Computer Vision (CV) (Dosovitskiy et al., 2020), Neural Machine Translation (NMT) (Chen et al., 2018), Document Summarization (Zhang et al., 2019; Pilault et al., 2020), and Protein Structure Prediction (Bahdanau et al., 2015). The main component of the Transformer model is an attention mechanism that identifies complex dependencies between tokens and efficiently captures tokens' correlation. However, standard self-attention suffers from quadratic memory and computation complexity, which arises from the $N \times N$ attention matrix, where $N$ is the sequence length. This problem is particularly significant during training, as it requires storing the attention matrix for gradient computation. Consequently, this significantly hinders the training of Transformer models with long sequences.

Recently, we have observed an increasing interest in training Transformer models with long sequences, especially when considering large language models (Scao et al., 2022; Zhang et al., 2022; Chowdhery et al., 2022). Various approaches address the quadratic memory issue in self-attention. One class of the methods is sparse-attention, which aims to perform only a subset of the attention computations while preserving the softmax function (Child et al., 2019; Zaheer et al., 2020). Another approach is Linearized Attention (LA), which replaces the softmax with a product of two functions (Choromanski et al., 2020; Katharopoulos et al., 2020). These methods reduce the computational and memory complexity of the attention mechanism while striving to maintain performance. One of LA's benefits is that it performs dense operations and does not require special HW or low-level implementation. However, despite their efficiency, LA methods often underperform compared to standard self-attention. Thus, understanding the reasons behind the superior performance of self-attention is crucial for designing an effective LA method.

In this paper, we propose a systematic way to develop an LA method with comparable performance to the Softmax Attention (SA). First, we define a holistic model of the SA and examine its characteristics. Then, we analyze the SA from three different perspectives, focusing on its statistical, informational, and algebraic properties. In particular, we characterize the probability distribution of the attention matrix and prove its log-normal nature. Moreover, we study the concentration behavior of the SA by analyzing its entropy and the spectral gap (Coste, 2017). Based on the proposed

model, we develop an LA method that emulates the distribution and concentration behavior of the SA, achieving comparable performance. Finally, we evaluate the effectiveness of our method on popular NLP benchmarks and compare it with other state-of-the-art methods. In summary, our contribution is as follows:

- We conduct an in-depth analysis of self-attention, characterizing its statistical, informational, and algebraic properties.

- Develop tools suitable for studying the concentration ability of the attention based on the entropy and the spectral gap metrics.

- Using the developed model and tools, we design Linear Log-Normal Attention (LLN Attention) with comparable performance to SA while requiring linear memory and computational complexity in the sequence length.

We have made the code of our method available for MindSpore[1] and PyTorch[2] frameworks.

## 2 BACKGROUND AND RELATED WORK

In this section, we present a brief overview of the attention mechanism and various methods for efficient and linearized attention. We review the most relevant works in the field, classifying them into different types of attention methods such as sparse attention, low-rank projection, memory-based, kernel-based approximations, and clustering-based methods.

### 2.1 BACKGROUND ON SELF-ATTENTION

In the seminal study of (Bahdanau et al., 2015), the authors proposed the attention mechanism, which was subsequently incorporated into the Transformer model (Vaswani et al., 2017). Since then, attention has become a fundamental building block for many Transformer-based models.

Consider a sequence of $N$ tokens, where each token represented by $d$-dimensional query, key, and value vectors, denoted as $\{\boldsymbol{q}_i\}_{i=1}^N$, $\{\boldsymbol{k}_i\}_{i=1}^N$, and $\{\boldsymbol{v}_i\}_{i=1}^N$, respectively. The SA is defined as:

$$\text{Attn}(\boldsymbol{q}_i, \{\boldsymbol{k}_1, \ldots, \boldsymbol{k}_N\}, \{\boldsymbol{v}_1, \ldots, \boldsymbol{v}_N\}) = \frac{\sum_{j=1}^N \kappa_{\exp}(\boldsymbol{q}_i, \boldsymbol{k}_j) \boldsymbol{v}_j^\top}{\sum_{l=1}^N \kappa_{\exp}(\boldsymbol{q}_i, \boldsymbol{k}_l)} \tag{1}$$

where $\kappa_{\exp}$ is an exponential kernel used in the softmax function:

$$\kappa_{\exp}(\boldsymbol{q}_i, \boldsymbol{k}_j) = \mathrm{e}^{\frac{\boldsymbol{q}_i^\top \boldsymbol{k}_j}{\sqrt{d}}} \tag{2}$$

The recent study by (Wright & Gonzalez, 2021) has examined SA from the perspective of the kernel method. Notably, the formulation of SA in eq. (1) can be seen as Nadaraya-Watson kernel regression (Nadaraya, 1964), where estimating some unknown function with joint distribution $p(\boldsymbol{k}, \boldsymbol{v})$ and density $p(\boldsymbol{k})$ with a kernel (Han et al., 2022). Moreover, as shown by (Tsai et al., 2019), other popular kernels, such as polynomial or Radial Basis Function (RBF), can be used instead of the exponential kernel. However, the performance may vary depending on the type of the kernel. A kernel method perspective of the attention allows us to address the problem of attention linearization by using the connection between any kernel and its feature embedding function $\Phi$, described by Mercer's theorem (Mercer, 1909):

$$\kappa(\boldsymbol{q}_i, \boldsymbol{k}_j) = \langle \Phi(\boldsymbol{q}_i), \Phi(\boldsymbol{k}_j) \rangle \tag{3}$$

---

[1] gitee.com/ynahshan/linear-lognormal-attention-ms
[2] github.com/ynahshan/linear-lognormal-attention

## 2.2 LINEARIZED ATTENTION

In recent years, several techniques have been proposed to address the quadratic cost associated with SA. Based on the taxonomy by (Zhu et al., 2021), these techniques can be categorized into five types: i) sparse attention mechanisms with predefined patterns, including sliding window approaches such as Sparse Transformer (Child et al., 2019), Axial Transformer (Ho et al., 2019), Blockwise Attention (Qiu et al., 2019), Longformer (Beltagy et al., 2020), and BigBird (Zaheer et al., 2020), where some of these works (Wang et al., 2021) manage to improve model convergence due to noise reduction; ii) low-rank projection methods, including Linformer (Wang et al., 2020), Synthesizer (Tay et al., 2020a), NystromFormer (Xiong et al., 2021), SkyFormer (Chen et al., 2021), and Cosformer (Qin et al., 2022b); iii) memory-based methods, such as Set Transformer (Lee et al., 2018b) and Compressive Transformers (Rae et al., 2019); iv) kernel-based approximation of the attention matrix, including Performer (Choromanski et al., 2020), Linear Transformers (Katharopoulos et al., 2020), and RFA (Peng et al., 2021); and v) similarity and clustering methods, including Reformer (Kitaev et al., 2020), Routing Transformer (Roy et al., 2020), Sinkhorn Attention (Tay et al., 2020b), and Clustered Attention (Vyas et al., 2020).

Some of these methods combine multiple types of efficient attention mechanisms. For instance, (Zhu et al., 2021) suggested a combination of low-rank projection and local window attention. Similarly, (Qin et al., 2022a) incorporate both kernel-based and block-wise attention in their approach. Our method also integrates kernel and block-wise techniques while suggesting a novel kernel approach that differs from that of (Qin et al., 2022a). By leveraging the benefits of multiple attention mechanisms, these techniques offer more efficient and accurate models for various NLP tasks.

Kernel-based attention requires selecting a feature embedding function $\Phi$ to compute the LA kernel eq. (3). Linearized attention can then be defined as:

$$\text{Attn}_{\text{lin}}(\boldsymbol{q}_i, \{\boldsymbol{k}_j\}_{j=1}^N, \{\boldsymbol{v}_j\}_{j=1}^N) = \frac{\sum_{j=1}^N \Phi_{\mathcal{Q}}(\boldsymbol{q}_i)^\top \Phi_{\mathcal{K}}(\boldsymbol{k}_j)}{\sum_{l=1}^N \Phi_{\mathcal{Q}}(\boldsymbol{q}_i)^\top \Phi_{\mathcal{K}}(\boldsymbol{k}_l)} \boldsymbol{v}_j^\top = \frac{\Phi_{\mathcal{Q}}(\boldsymbol{q}_i)^\top \sum_{j=1}^N \Phi_{\mathcal{K}}(\boldsymbol{k}_j) \boldsymbol{v}_j^\top}{\Phi_{\mathcal{Q}}(\boldsymbol{q}_i)^\top \sum_{l=1}^N \Phi_{\mathcal{K}}(\boldsymbol{k}_l)} \quad (4)$$

The choice of feature embedding function is crucial, as we demonstrate later in section 4. Different works suggest different types of embedding functions for this purpose. For example, Performer (Choromanski et al., 2020) uses an exponential function, Linear Transformers (Katharopoulos et al., 2020) uses the ELU function, and RFA (Peng et al., 2021) uses trigonometric functions to approximate the Gaussian kernel with Fourier features. However, none of these works have analyzed the properties of the attention mechanism induced by these functions.

## 3 DISSECTING SOFTMAX ATTENTION

In the previous section, we discussed the LA concept. Although this concept may seem straightforward, creating an LA mechanism that effectively handles complex tasks presents a challenging problem. Typically, the LA of the form in eq. (4) performs worse than the SA. To gain insight into the superiority of the SA, we conduct a thorough analysis of its properties. We start by characterizing the distribution of the attention matrix since it is a core element of the attention mechanism. Then, we study the connection between its entropy, spectral gap, and concentration ability of the self-attention.

We begin our analysis by formalizing a model based on the SA from Equation (1). Our model assumes that queries and keys approximately follow a Gaussian distribution. This assumption is reasonable due to the Central Limit Theorem (CLT) (Lee et al., 2018a) and accepted in literature for tractability purposes (Ioffe & Szegedy, 2015; Banner et al., 2018). Moreover, let us assume the mean of queries and keys is approximately zero, which is a valid assumption due to the layer normalization presence in the Transformer models.

**Model Definition.** *Let $\boldsymbol{q}_i, \boldsymbol{k}_j \in \mathbb{R}^d$ be a Gaussian vectors, where elements $q_{i\ell} \sim \mathcal{N}(0, \sigma_q^2)$ and $k_{j\ell} \sim \mathcal{N}(0, \sigma_k^2)$, $\forall l$. Let $a_{ij} = \boldsymbol{q}_i^\top \boldsymbol{k}_j / \sqrt{d}$ the attention score of pair $i, j$, whose variance can be expressed as $\sigma_{\text{sm}}^2 = \sigma_q^2 \sigma_k^2 + C_{\text{cross}}$, where $C_{\text{cross}}$ is the cross-covariance of the squared queries and keys (Goodman, 1960). We define a temperature of the SA as:*

$$\tau_{\mathrm{sm}} = \frac{1}{\sigma_{\mathrm{sm}}} = \frac{1}{\sqrt{\sigma_q^2 \sigma_k^2 + C_{\mathrm{cross}}}} \tag{5}$$

*Denote $\tilde{a}_{ij} = a_{ij} \, / \, \sigma_{\mathrm{sm}}$ and let $\boldsymbol{P}^{(SM)} \in \mathbb{R}^{N \times N}$ the SA matrix, where $N$ is sequence length such that:*

$$P_{ij}^{(SM)} = \frac{e^{\tilde{a}_{ij}/\tau_{\mathrm{sm}}}}{\sum_{l=1}^{N} e^{\tilde{a}_{il}/\tau_{\mathrm{sm}}}} \tag{6}$$

The form in Equation (6) is significant as it demonstrates the connection between SA and implicit temperature parameter imposed by the variance of attention inputs. We can draw an analogy between the SA training and stochastic processes, where controlling the temperature allows balancing between exploration and exploitation. High temperature results in equal probabilities for all tokens (exploration), whereas low-temperature results in a high probability for one or few tokens, emphasizing it (exploitation of this particular token).

### 3.1 Characterizing distribution of Softmax Attention

Let us now characterize the probability distribution of the SA. By analyzing its probability distribution, we can gain valuable insights into the behavior of the SA and reveal its statistical properties. In particular, the distribution of $\boldsymbol{P}^{(SM)}$ plays a crucial role in quantifying the variability of its entries. This variability is closely related to the concentration ability of the SA, a topic that we will explore in more detail in subsequent sections.

**Proposition 3.1.** *Let $\boldsymbol{q}$ and $\boldsymbol{k}$ be Gaussian vectors, where $q_i \sim \mathcal{N}(0, \sigma_q^2)$ and $k_j \sim \mathcal{N}(0, \sigma_k^2)$, $\forall i, j$. Then, for moderate values of $\sigma_q^2, \sigma_k^2$ and large enough $N$ the distribution of $\boldsymbol{P}^{(SM)}$ can be approximated by a log-normal distribution with parameters $\mu_{\mathrm{sm}} = -\ln N - \frac{1}{2}\sigma_{\mathrm{sm}}^2$ and $\sigma_{\mathrm{sm}}^2 = \sigma_q^2 \sigma_k^2 + C_{\mathrm{cross}}$.*

The key behind the proof is to approximate the denominator in eq. (6) with log-normal distribution by (Fenton, 1960) theorem. Then, since the numerator is also log-normal by the CLT, the resulting ratio can be approximated by a log-normal distribution. It leads to the log-normal distribution of the $\boldsymbol{P}^{(SM)}$. The detailed proof is given in Appendix A.5.

The log-normal probability distribution of the SA matrix helps us understand the attention mechanism. The skewness of log-normal distribution emphasizes certain positions and enables concentration. The temperature parameter controls uncertainty, influencing the balance between exploration and exploitation during training.

### 3.2 Analyzing Self-Attention through Markov Chain Perspective

To delve deeper into the self-attention mechanism, we draw inspiration from the principles of Markov chains (Levin et al., 2006). Each row of the self-attention matrix represents the correlation between a specific token and all other tokens. These correlations closely resemble transition probabilities in classical Markov chains. The self-attention matrix continuously evolves during training, eventually converging to a final model.

#### 3.2.1 Entropy and Attention Concentration

A crucial parameter in understanding such a stochastic system is its entropy, a metric commonly used to measure the uncertainty or randomness associated with the state transitions of a Markov chain. In the context of self-attention, entropy serves as a valuable tool to evaluate the concentration ability of the self-attention. We refer to this as Attention Concentration (AC), which essentially measures the model's ability to direct its focus toward specific tokens, thereby extracting relevant information from the input sequence. Previous studies (Ghader & Monz, 2017; Vig & Belinkov, 2019) have proposed using entropy to measure the AC. Lower entropy indicates a greater focus on a

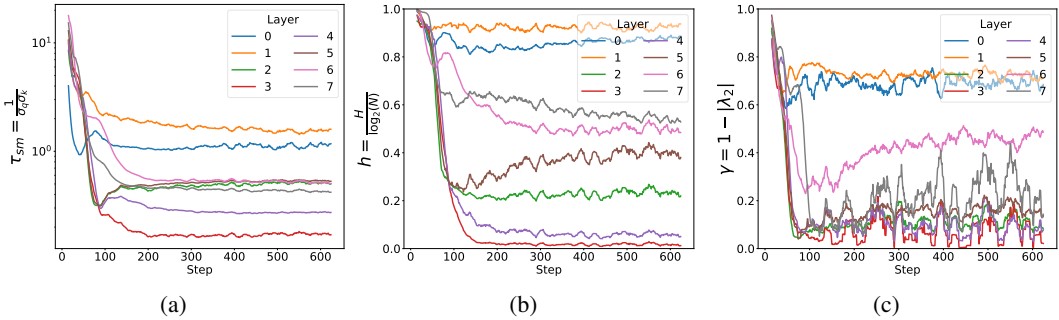

Figure 1: Temperature (left), entropy (center), and spectral gap (right) during training of the small RoBERTa model with a single head per layer in every training step (X-axis).

few tokens, while higher entropy indicates more uniformly distributed attention. To formally define the entropy of the attention matrix $\boldsymbol{P}^{(\mathrm{SM})}$ we average the entropy of individual rows as following:

$$H(\boldsymbol{P}^{(\mathrm{SM})}) = -\frac{1}{N}\sum_{i=1}^{N}\sum_{j=1}^{N} P_{ij}^{(\mathrm{SM})} \log_2(P_{ij}^{(\mathrm{SM})}) \tag{7}$$

Note that by eq. (6), the attention matrix can be represented in terms of its temperature. To further explore the connection between the AC and temperature, we present the following theorem, which characterizes the relationship between the entropy of the SA and its temperature:

**Theorem 3.2.** *The entropy in eq. (7) is monotonically increasing with temperature $\tau_{\mathrm{sm}}$.*

To prove the theorem, we consider the derivative of the entropy with respect to the temperature and show it is always positive. For detailed proof, refer to Appendix A.4.1.

According to Theorem 3.2, the entropy of the SA increases with the temperature, which controls the concentration of the SA. Essentially, a higher temperature results in a more dispersed distribution of attention. Conversely, a lower temperature makes it easier to focus on specific tokens. Moreover, the temperature controls the exploration (higher entropy) and exploitation (lower entropy) of the states within the chain. Figure 1a shows how temperature decreases during training, resulting in a more confident state (lower entropy) Figure 1b. Notably, while the first layers of the model retain high entropy, allowing exploration, the entropy of the middle layers decreases and becomes approximately zero, leading to exploitation in those layers.

### 3.2.2 SPECTRAL GAP AND ATTENTION CONCENTRATION

In the analysis of Markov chains, the spectral gap is a valuable metric to consider because it provides insights into the speed at which the chain reaches its stationary state (Coste, 2017). In other words, it quantifies the rate of convergence, where the larger values of the spectral gap indicate a faster convergence process, while a smaller one suggests a slower one. When applied to attention mechanisms, the spectral gap can provide insights into the rate at which the attention mechanism focuses on specific elements within the input sequence. Including the spectral gap in our analysis allows us a more comprehensive understanding of the SA mechanism from an algebraic perspective.

The spectral gap measures the difference between the first and the second largest eigenvalues. Since the attention matrix is a *stochastic matrix* (Meyer, 2000), it follows from the Perron-Frobenius theorem (Samelson, 1957) that its largest eigenvalue is $\lambda_1 = 1$. Therefore, the spectral gap is $\gamma = 1 - |\lambda_2|$, where $\lambda_2$ is the second largest eigenvalue. In Theorem 3.3, we establish a relationship between the variance of the attention matrix and the spectral gap.

**Theorem 3.3.** *Let $\boldsymbol{P} \in \mathbb{R}^{N \times N}$ right stochastic matrix with eigenvalues $\lambda_1, \ldots, \lambda_N$ ordered by their absolute values, where $\lambda_1 = 1 \geq |\lambda_2| \geq \cdots \geq |\lambda_N|$. Let $\bar{\boldsymbol{v}}_{max}$ be the major principal component of the centered version of $\boldsymbol{P}$. Then, $\lambda_2^2 = \sigma_{\bar{\boldsymbol{v}}_{max}}^2$, where $\sigma_{\bar{\boldsymbol{v}}_{max}}^2$ represents the amount of variance in the direction specified by the major principal component $\bar{\boldsymbol{v}}_{max}$.*

To prove this theorem, we deflate the $\lambda_1$ of the attention matrix and express the variance in the direction of the major principal component. The detailed proof is provided in Appendix A.4.2.

**Theorem 3.4.** *The variance of the attention matrix $\boldsymbol{P}^{(SM)}$ is monotonically decreasing with temperature $\tau_{\mathrm{sm}}$.*

To prove this theorem, we consider the derivative of the variance with respect to the temperature and show it is always negative. For detailed proof, refer to Theorem A.3.

According to Theorem 3.3, the magnitude of variability in the direction of the major principal component $\bar{\boldsymbol{v}}_{\mathrm{max}}$ is equal to $\lambda_2$, consequently, the spectral gap increases as the variability decreases. Together with the Theorem 3.4, we can conclude that the spectral gap increases with the temperature, similarly to the entropy. However, biasing the stochastic matrix towards a particular column also affects the variability. When $\boldsymbol{P}$ is biased toward a specific column, the variability within the columns decreases, resulting in a smaller value of $\lambda_2$ and a higher spectral gap, regardless of the temperature. Therefore, we can conclude that the spectral gap only increases with temperature when the attention matrix is unbiased. This phenomenon led us to refer to the spectral gap as a measure of *Unbiased Attention Concentration*. Figure 1c depicts the change in the spectral gap during training. In most layers, the spectral gap decreases during training, indicating improved AC. However, in some layers, the spectral gap increases while the temperature remains constant, suggesting that the attention matrix is biased. This observation justifies that the spectral gap carries additional information to entropy.

## 4 DESIGN OF LINEARIZED ATTENTION

In the previous section, we presented a holistic model of SA and conducted a thorough analysis of its properties. Specifically, we identified the log-normal distribution of the SA matrix. Additionally, we analyzed concentration behavior dictated by the temperature parameter. We can measure AC using the entropy (biased) and the spectral gap (unbiased) metrics. In this section, we design the LA method based on the defined model, which resembles similar characteristics and imitates SA behavior. In particular, our LA model should have log-normal distribution with similar moments. Moreover, it should emulate the concentration pattern of the SA by matching its entropy and spectral gap curves. As a result, we expect our LA method to achieve performance comparable to the SA.

### 4.1 LINEAR LOG-NORMAL ATTENTION

Designing LA according to Equation (4) requires selecting a feature embedding function $\Phi$, a core element of this attention. The choice of this function has a crucial effect on the LA performance. According to our model, we start by satisfying the log-normality requirement, as most functions do not have this property. For example, the Rectified Linear Unit (ReLU) can not produce log-normal distribution as being almost linear. On the other hand, the exponential function induces log-normal distribution for Gaussian inputs, which justifies its selection as a feature embedding function $\Phi$. However, to match the concentration behavior of the SA, we must force the LA to produce similar entropy and spectral gap curves with respect to the temperature as in SA. To achieve this goal, we introduce additional parameters, which we tune to perform moment matching between the LA distribution and that of the SA.

Accordingly, let us denote by $\Phi_{\mathcal{Q}}(\boldsymbol{q}) = e^{\alpha \boldsymbol{q}}$ and $\Phi_{\mathcal{K}}(\boldsymbol{k}) = e^{\beta \boldsymbol{k}}$ the feature embedding functions, where $\alpha, \beta \in \mathbb{R}^+$ are hyper-parameters that must be carefully selected to ensure our LA closely approximates the SA. We define the Linear Log-Normal (LLN) Attention as:

$$\mathrm{Attn}_{\mathrm{LLN}}(\boldsymbol{q}_i, \{\boldsymbol{k}_1, \ldots, \boldsymbol{k}_N\}, \{\boldsymbol{v}_1, \ldots, \boldsymbol{v}_N\}) = \frac{\sum_{j=1}^{N} e^{\alpha \boldsymbol{q}_i^\top} e^{\beta \boldsymbol{k}_j}}{\sum_{l=1}^{N} e^{\alpha \boldsymbol{q}_i^\top} e^{\beta \boldsymbol{k}_l}} \boldsymbol{v}_j^\top \tag{8}$$

Where each entry of the LLN attention matrix can be expressed as:

$$P_{ij}^{(\mathrm{LLN})} = \frac{e^{\alpha \boldsymbol{q}_i^\top} e^{\beta \boldsymbol{k}_j}}{\sum_{l=1}^{N} e^{\alpha \boldsymbol{q}_i^\top} e^{\beta \boldsymbol{k}_l}} \tag{9}$$

Similarly to the analysis of the SA model, we assume zero mean of queries and keys. Then, to show that the LLN Attention matrix follows a log-normal distribution, we prove the following:

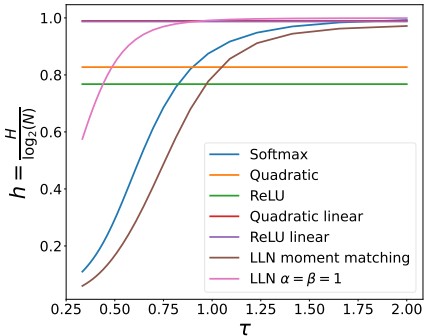 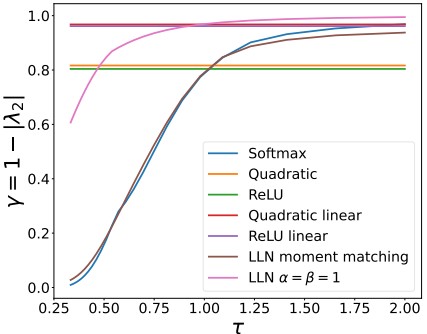

Figure 2: Comparison of entropy (left) and spectral gap (right) for various types of attention kernels. The figure shows that the entropy and spectral gap of the LLN Attention with the moment matching is similar to those of the SA.

**Proposition 4.1.** *Let $q$ and $k$ be Gaussian vectors, where $q_i \sim \mathcal{N}(0, \sigma_q^2)$ and $k_j \sim \mathcal{N}(0, \sigma_k^2)$, $\forall i, j$. Then, for moderate values of $\sigma_q^2$ and $\sigma_k^2$, the distribution of $\boldsymbol{P}^{(LLN)}$ can be approximated by a log-normal distribution with variance $\sigma_{lln}^2 = a \cdot (\alpha^2 \sigma_q^2 + \beta^2 \sigma_k^2) + b$, where $a$ and $b$ are constants.*

The main steps of the proof are approximating the numerator and denominator in eq. (9) using the log-normal distribution, following the theorem by (Fenton, 1960). Then, split the analysis into three cases to express the variance, as suggested by (Romeo et al., 2003). The detailed proof is given in Appendix A.6.

Further, we have to ensure the concentration behavior of the LLN Attention is similar to that of the SA. To that end, it is necessary to determine appropriate values for the hyperparameters $\alpha$ and $\beta$. In the following, we estimate these parameters by performing moment matching to the distribution of the SA. Since the log-normal distribution is parameterized only by the first and second moments, we can align the LLN Attention distribution with the SA by ensuring equivalence of the first two moments.

Interestingly, Proposition 4.1 implies linear dependency between the variance of queries and keys and $\sigma_{lln}^2$. This linear connection facilitates the calculation of constants $a$ and $b$. It allows the application of linear interpolation on the randomly generated Gaussian samples $q$ and $k$ to perform the moment matching between LLN and SA. We provide a detailed description of the technical aspects of this moment-matching technique in Appendix A.7.

Finally, by requiring $\sigma_{lln} = \sigma_{sm}$ and expressing it in terms of $\alpha$ and $\beta$, we can determine $a$ and $b$ parameters. We point out that there is no closed analytical solution for which both the mean and variance of LLN and SA align. Yet, since the concentration is mostly affected by the variance of the attention matrix, we only match the variances of the LLN and SA. Further, to simplify the solution, we also let $\alpha^2 \sigma_q^2 = \beta^2 \sigma_k^2 = \frac{1}{2}\tilde{\sigma}^2$. Hence, we obtain the following:

$$\alpha = \frac{\tilde{\sigma}}{\sqrt{2}\sigma_q}; \quad \beta = \frac{\tilde{\sigma}}{\sqrt{2}\sigma_k}; \quad \tilde{\sigma} = \sqrt{\frac{1}{a}(\sigma_q^2 \sigma_k^2 - b)} \tag{10}$$

A detailed derivation of Equation (10) is given in Appendix A.7. Note that, like in the SA, we can introduce a temperature parameter of the LLN Attention that controls the concentration. Specifically, let us define the temperature of the LLN Attention to be:

$$\tau_{lln} = \frac{1}{\sqrt{a \cdot (\alpha^2 \sigma_q^2 + \beta^2 \sigma_k^2) + b}} \tag{11}$$

In Figure 2, we demonstrate that the moment matching is essential to align the entropy and the spectral gap of the LLN Attention with those of the SA to achieve the required concentration. Moreover, other popular kernels, such as quadratic, ReLU, and their linear counterparts, are indifferent to the temperature, which may result in poor concentration and potentially degraded performance. In conclusion, LLN Attention satisfies the desired log-normal distribution property and concentration behavior required by the SA model. Therefore, it should achieve comparable results to the SA.

## 4.2 THE OVERALL ARCHITECTURE

In this section, we present the experimental results of the LLN Attention method on NLP tasks, while more experiments on Image Classification and LRA(Tay et al., 2020c) benchmark available in the Appendix A.8.

The LLN Attention effectively scales to long sequences while maintaining high concentration, allowing capturing long-range interactions. However, for short-range connections, it may be less effective. Recently, a study by (Qin et al., 2022a) emphasized the attention dilution issue of LA methods. Specifically, LA may overlook neighboring structures, leading to the "dilution" of short-range interactions. To address this issue, the authors proposed a hybrid approach that combines LA with block-

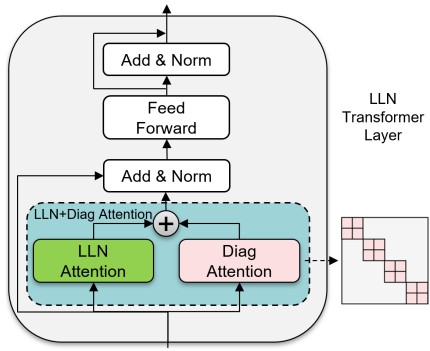

Figure 3: LLN Transformer layer architecture.

diagonal attention, which retains the $O(N)$ memory and computational complexity of LA. This block-diagonal attention is a regular SA applied on smaller pieces of the input, computing only the diagonal of the original attention matrix. Such block-diagonal attention can not scale to longer sequences, but it is useful to improve the performance of the LA method.

We incorporate this technique into LLN Attention, combining the LLN and block-diagonal attention into a unified layer by averaging the outputs of both components Figure 3. While the block-diagonal mechanism effectively captures short-range interactions within its confined block scope, LLN excels in catching broader, long-range connections. This combined approach enhances the performance of LLN Attention and stabilizes training by reducing the magnitude of the gradients (Qin et al., 2022a).

## 5 EXPERIMENTS

We first pre-train the bidirectional RoBERTa encoder model (Liu et al., 2019) using LLN Attention on the WikiText-103 corpus (Merity et al., 2018). During pre-training, we monitor the convergence of the model and compare its performance to the SA model. In Appendix A.8.1, we show that the loss of the LLN Attention closely follows the loss of the SA, indicating similar convergence behavior.

Next, to evaluate the performance of LLN Attention on downstream tasks, we fine-tune our pre-trained model on several NLP tasks from the General Language Understanding Evaluation (GLUE) dataset (Wang et al., 2018). These tasks include Multi-Genre Natural Language Inference (MNLI), Question-answering Natural Language Inference (QNLI), Quora Question Pairs (QQP), and Stanford Sentiment Treebank (SST-2). For all our experiments, we use the Fairseq framework (Ott et al., 2019) with the default configuration and hyperparameters of the RoBERTa-base model.[3]

Table 1 provides a detailed comparison of the accuracy achieved by each method on each task. The LLN Attention method outperforms the other LA methods with an average accuracy of 86.9%. These results confirm the superior capability of LLN Attention in achieving competitive performance with SA on a range of NLP tasks.

### 5.1 SPEED AND MEMORY CONSUMPTION

In this section, we evaluate the training time and memory usage of the LLN Attention, comparing it to the SA and Nyströmformer, which outperform most of the LA methods available. In the comparison, we used the RoBERTa-base model with a batch size of one and performed all measurements on a commodity GPU.

The results in Table 2 confirm that LLN Attention scales linearly with sequence length, as expected, and can handle at least four times longer sequences than SA. Moreover, the LLN Attention method

---

[3]https://github.com/facebookresearch/fairseq/blob/main/examples/roberta/README.md

| Method | MNLI | QNLI | QQP | SST-2 | Avg ↑ |
|---|---|---|---|---|---|
| SA baseline (Bahdanau et al., 2015) | 80.3 | 87.2 | 89.9 | 90.6 | 87.0 |
| Reformer (Kitaev et al., 2020) | 35.4 | - | 63.2 | 50.9 | 49.8 |
| Performer (Choromanski et al., 2020) | 58.8 | 63.4 | 79.1 | 81.4 | 70.6 |
| ELU (Katharopoulos et al., 2020) | 74.8 | 82.5 | 86.9 | 87.2 | 82.8 |
| Longformer (Beltagy et al., 2020) | 77.2 | - | 85.5 | 88.6 | 83.7 |
| Transformer LS (Zhu et al., 2021) | 77.0 | 84.8 | 86.8 | 90.2 | 84.7 |
| TNN (Qin et al., 2023) | 76.72 | 85.06 | 88.3 | 90.6 | 85.17 |
| T2 (Qin et al., 2022a) | 77.28 | 85.39 | 88.56 | 90.71 | 85.48 |
| CosFormer (Qin et al., 2022b) | 76.7 | - | 89.2 | 91 | 85.6 |
| T1 (Qin et al., 2022a) | 79.06 | 87.0 | 88.61 | 91.17 | 86.46 |
| Flash (Hua et al., 2022) | 79.45 | **87.1** | 88.83 | 90.71 | 86.52 |
| Nyströmformer* (Xiong et al., 2021) | 80.9(-1.5) | 88.7(-1.6) | 86.3(-1.) | 91.4(+1.4) | 86.8(-0.7) |
| LLN Attention (Ours) | 77.0 | 85.1 | 88.9 | 90.6 | 85.4 |
| LLN+Diag Attention (Ours) | **80.0** | 86.5 | **89.7** | **91.6** | **86.9** |

Table 1: Accuracy achieved by various LA methods on multiple NLP tasks from the GLUE dataset, including MNLI, QNLI, QQP, and SST-2. The results of Nyströmformer are given in (Xiong et al., 2021), while the results of the rest are given in (Qin et al., 2022a). Note that for methods marked with *, which have a different baseline in the original paper, we also provide an accuracy drop in ().

| | | sequence length | | | | | |
|---|---|---|---|---|---|---|---|
| | Method | 512 | 1024 | 2048 | 4096 | 8192 | 16384 |
| | Softmax Attention | 4. | 5.5 | 12.6 | 32.1 | OOM | OOM |
| Memory [GB] | Nyströmformer | 4. | 4.5 | 5.5 | 7.3 | 11.6 | 19.1 |
| | LLN Attention | 4.1 | 4.4 | 5.7 | 7.5 | 12. | 20.1 |
| | LLN+Diag Attention | 4.1 | 4.6 | 6.1 | 8.1 | 13.4 | 23. |
| | Softmax Attention | 0.95 | 1.05 | 2.4 | 6.8 | OOM | OOM |
| Time [sec/it] | Nyströmformer | 1.8 | 1.9 | 2.6 | 4.7 | 8.8 | 16.7 |
| | LLN Attention | 1. | 1.05 | 1.6 | 3.2 | 6.1 | 11.8 |
| | LLN+Diag Attention | 1.2 | 1.3 | 1.9 | 3.6 | 6.9 | 13.3 |

Table 2: Memory usage and training time per iteration of SA, Nyströmformer, LLN, and LLN+Diag on RoBERTa model, with varying sequence lengths. "OOM" indicates an "Out Of Memory" error.

requires nearly the same amount of memory as Nyströmformer, with Diag Attention adding only a 10% memory overhead to the LLN Attention. Notably, both LLN and LLN+Diag Attention demonstrate superior speed compared to Nyströmformer.

## 6 CONCLUSION

In this paper, we introduced a novel LLN Attention method that incorporates the essential properties of the SA, such as the log-normal distribution of the attention matrix and its concentration behavior, while offering linear time and memory complexity. Our approach includes a moment-matching technique to match the attention matrix's log-normal distribution with that of the SA, resulting in improved attention concentration and model performance. In addition, we conducted a comprehensive analysis of the SA, characterizing its distribution and suggesting entropy and the spectral gap metrics for attention concentration analysis. To the best of our knowledge, this is the first work to study self-attention from this perspective. Finally, our experimental results demonstrated that LLN Attention outperforms many existing LA methods on several NLP tasks, demonstrating its competitiveness and potential to enhance attention performance on long sequences. Overall, our contribution provides a foundation for future research and improvements in attention mechanisms.

## ACKNOWLEDGEMENTS

We extend our gratitude to Dr. Eliezer Levy, Dror Mizrachi, Dr. Su Teng, Wang ShengNan, and Dror Meirovich for their valuable support and fruitful discussions.

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

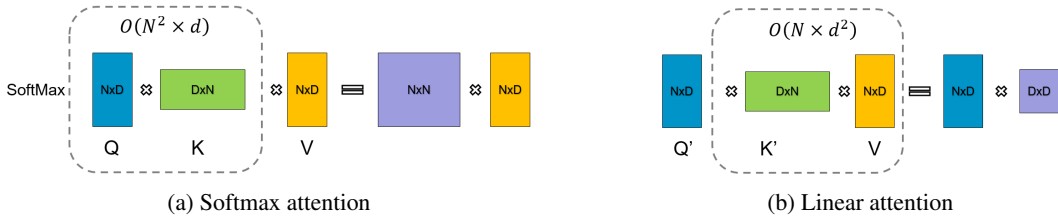

(a) Softmax attention             (b) Linear attention

Figure 4: A block diagram of the computational complexity for the Softmax Attention and Linearized Attention.

# A   APPENDIX

## A.1  FORMULATION OF THE SELF-ATTENTION (SA)

One of the most widely used variants of the self-attention mechanism is the scaled dot-product attention (Vaswani et al., 2017). In this formulation, the input vectors $[\boldsymbol{x}_1, \ldots, \boldsymbol{x}_N]^\top := \boldsymbol{X} \in \mathbb{R}^{N \times d}$ are projected into query, key, and value vectors as follows:

$$
\begin{aligned}
[\boldsymbol{q}_1, \ldots, \boldsymbol{q}_N]^\top := \boldsymbol{Q} = \boldsymbol{X}\boldsymbol{W}_q^\top \in \mathbb{R}^{N \times d} \\
[\boldsymbol{k}_1, \ldots, \boldsymbol{k}_N]^\top := \boldsymbol{K} = \boldsymbol{X}\boldsymbol{W}_k^\top \in \mathbb{R}^{N \times d} \\
[\boldsymbol{v}_1, \ldots, \boldsymbol{v}_N]^\top := \boldsymbol{V} = \boldsymbol{X}\boldsymbol{W}_v^\top \in \mathbb{R}^{N \times d}
\end{aligned}
\tag{12}
$$

Here, $\boldsymbol{W}_q, \boldsymbol{W}_k, \boldsymbol{W}_v \in \mathbb{R}^{d \times d}$ are learnable parameter matrices. The attention function is then computed on the query, key, and value vectors using the following equation:

$$
\text{Attn}(\boldsymbol{q}_i, \{\boldsymbol{k}_1, \ldots, \boldsymbol{k}_N\}, \{\boldsymbol{v}_1, \ldots, \boldsymbol{v}_N\}) = \sum_{j=1}^{N} \text{softmax}\left(\frac{\boldsymbol{q}_i^\top \boldsymbol{k}_j}{\sqrt{d}}\right) \boldsymbol{v}_j^\top
\tag{13}
$$

The dot-product term is scaled by the factor $\frac{1}{\sqrt{d}}$ to ensure the stability of computations. This scaling factor accounts for the variance of the dot-product $\boldsymbol{q}_i^\top \boldsymbol{k}_j$, which grows with the dimensionality $d$. In this paper, we refer to this scaled dot-product attention as "softmax attention" or "standard attention," which is defined in eq. (1).

## A.2  LINEARIZED ATTENTION (LA)

Let $\boldsymbol{Q}, \boldsymbol{K}, \boldsymbol{V} \in \mathbb{R}^{N \times d}$ denote the queries, keys, and values of the attention mechanism. Computing softmax-attention with Equation equation 13 requires calculating the quadratic matrix $\boldsymbol{Q}\boldsymbol{K}^\top = \left[\boldsymbol{q}_i^\top \boldsymbol{k}j\right]_{N \times N}$, which has a complexity of $\mathcal{O}(N^2)$ with respect to the sequence length $N$ as illustrated in Figure 4a.

Alternatively, if we can decompose $\text{softmax}\left(\boldsymbol{Q}\boldsymbol{K}^\top\right)$ into $\boldsymbol{Q}'\boldsymbol{K}'^\top$, we can use the associativity property of matrix multiplication to compute $\boldsymbol{Q}'\left(\boldsymbol{K}'^\top\boldsymbol{V}\right)$ from right to left. Such a decomposed computation has a linear complexity in the sequence length $N$, as illustrated in Figure 4b. To obtain this decomposition, the un-normalized matrix $e^{\boldsymbol{Q}\boldsymbol{K}^\top}$ is replaced with matrix $\Phi(\boldsymbol{Q})\Phi(\boldsymbol{K})^\top$, where $\Phi$ is a feature map that is applied row-wise, i.e., $\boldsymbol{Q}' = \Phi(\boldsymbol{Q})$ and $\boldsymbol{K}' = \Phi(\boldsymbol{K})$. This form of linear attention can be expressed as follows:

$$
\text{Attn}_{\text{lin}}(\boldsymbol{q}_i, \{\boldsymbol{k}_1, \ldots, \boldsymbol{k}_N\}, \{\boldsymbol{v}_1, \ldots, \boldsymbol{v}_N\}) = \frac{\sum_{j=1}^{N} \Phi(\boldsymbol{q}_i)^\top \Phi(\boldsymbol{k}_j)}{\sum_{l=1}^{N} \Phi(\boldsymbol{q}_i)^\top \Phi(\boldsymbol{k}_l)} \boldsymbol{v}_j^\top = \frac{\Phi(\boldsymbol{q}_i)^\top \sum_{j=1}^{N} \Phi(\boldsymbol{k}_j)\boldsymbol{v}_j^\top}{\Phi(\boldsymbol{q}_i)^\top \sum_{l=1}^{N} \Phi(\boldsymbol{k}_l)}
\tag{14}
$$

Due to aggregation over $N$ elements in both the numerator and denominator, the memory complexity of this linearized attention mechanism is linear with respect to the sequence length $N$.

## A.3 KERNEL VIEW OF THE SELF-ATTENTION

The SA mechanism can be viewed as a Nadaraya-Watson kernel regression (Nadaraya, 1964) as shown by Wright & Gonzalez (2021). Specifically, the kernel density estimation (KDE) of some unknown function with joint distribution $p(\boldsymbol{k}, \boldsymbol{v})$ and density $p(\boldsymbol{k})$ with a kernel $\kappa$ (Han et al., 2022). Therefore, the kernel function can be used to express the self-attention mechanism as follows:

$$\text{Attn}(\boldsymbol{q}_i, \{\boldsymbol{k}_1, \ldots, \boldsymbol{k}_N\}, \{\boldsymbol{v}_1, \ldots, \boldsymbol{v}_N\}) = \frac{\sum_{j=1}^N \kappa(\boldsymbol{q}_i, \boldsymbol{k}_j)\boldsymbol{v}_j^\top}{\sum_{l=1}^N \kappa(\boldsymbol{q}_i, \boldsymbol{k}_l)}. \tag{15}$$

he KDE form of Equation (15) is a generalization of softmax attention in Equation (13) where we use a softmax kernel

$$\kappa_{\text{sm}}(\boldsymbol{q}_i, \boldsymbol{k}_j) = e^{\frac{\langle \boldsymbol{q}_i, \boldsymbol{k}_j \rangle}{\sqrt{d}}} \tag{16}$$

Interestingly, the kernel view of self-attention reveals that the functionality of the attention mechanism remains intact even when the kernel is modified. However, the performance of the attention mechanism can vary depending on the choice of the kernel function, as demonstrated by Tsai et al. (2019). It emphasizes the significance of considering different kernels and their impact on the performance of linearized attention. Additionally, the kernel view provides a valuable perspective for tackling the linearization problem within the framework of the kernel method.

According to Mercer's theorem (Mercer, 1909), any positive-definite kernel (Mercer kernel) can be represented as an inner product of symmetric features. Let $\mathcal{X} \in \mathbb{R}^{N \times d}$, $\boldsymbol{x}_1, ..., \boldsymbol{x}_N \in \mathcal{X}$ with kernel function $\kappa : \mathcal{X} \times \mathcal{X} \to \mathbb{R}$, then

$$\kappa(\boldsymbol{x}_i, \boldsymbol{x}_j) = \langle \Phi(\boldsymbol{x}_i), \Phi(\boldsymbol{x}_j) \rangle_{\mathcal{F}_\mathcal{H}} \tag{17}$$

where $\Phi : \mathcal{X} \to \mathcal{F}_\mathcal{H}$ is a function mapping the inputs to a Hilbert space of feature functions $\mathcal{F}_\mathcal{H}$. However, the dimensionality of $\mathcal{F}_\mathcal{H}$ can be large or infinite, making explicit computation of the features infeasible. This is where the kernel trick comes in: the kernel function can be computed without explicitly computing the features. However, storing the attention matrix still requires $\mathcal{O}(N^2)$ memory complexity. To address this issue, we can design a kernel function such that the dimensionality of $\mathcal{F}_\mathcal{H}$ is much smaller than $N$, which allows for $\mathcal{O}(N)$ memory complexity. For example, consider $\Phi_\mathcal{Q} : \mathcal{Q} \to \mathbb{R}^d$ and $\Phi_\mathcal{K} : \mathcal{K} \to \mathbb{R}^d$ for some $d \ll N$, the attention can be computed using the following linear kernel function:

$$\kappa(\boldsymbol{q}_i, \boldsymbol{k}_j) = \langle \Phi_\mathcal{Q}(\boldsymbol{q}_i), \Phi_\mathcal{K}(\boldsymbol{k}_j) \rangle \tag{18}$$

Consequently, using the associativity property described in section A.2 can be used to compute this kernel function with $\mathcal{O}(N)$ memory complexity.

## A.4 ANALYSIS OF THE ENTROPY, VARIANCE AND SPECTRAL GAP OF THE SOFTMAX ATTENTION MARIX

We start our analysis by providing additional definitions which extend those in Section 3 and proving a couple of lemmas. Denote by $\boldsymbol{a} \in \mathbb{R}^N$ a single row of the attention scores matrix $\boldsymbol{A}$. By denoting a corresponding row of the normalized attention scores matrix $\tilde{\boldsymbol{A}}$ by $\boldsymbol{x}$, such that $\tau \boldsymbol{a} = \boldsymbol{x}$, we can write $\boldsymbol{p} \in \mathbb{R}^N$ a single row of the softmax attention matrix from eq. (6) as:

$$\boldsymbol{p} = \text{softmax}(\boldsymbol{x}, \tau) = \frac{e^{\frac{\boldsymbol{x}}{\tau}}}{\sum_{j=1}^N e^{\frac{x_j}{\tau}}} \tag{19}$$

Where $\tau$ is the temperature of SA from eq. (5). Similarly to eq. (7), we define the entropy of a single row $\boldsymbol{p}$ as:

$$H(\boldsymbol{p}) = -\sum_{i=1}^N p_i \log_2(p_i) \tag{20}$$

Additionally, we define a variance of the single row $\boldsymbol{p}$ of the SA matrix as:

$$\sigma_{\boldsymbol{p}}^2 = \frac{1}{N} \sum_{i=1}^{N} \left( p_i - \frac{1}{N} \sum_{j=1}^{N} p_j \right)^2 = \frac{1}{N} \sum_{i=1}^{N} \left( p_i - \frac{1}{N} \right)^2 \tag{21}$$

Denote $\mu = \sum_{i=1}^{N} p_i x_i$

**Lemma A.1.** *The following holds* $\sum_{i=1}^{N} p_i (x_i - \mu) = 0$

*Proof.*

$$\sum_{i=1}^{N} p_i (x_i - \mu) = \sum_{i=1}^{N} p_i x_i - \mu \underbrace{\sum_{i=1}^{N} p_i}_{1} = \mu - \mu = 0$$

$\square$

**Lemma A.2.** *Let $\boldsymbol{p}$ as in eq. (19), then the following holds.*

$$\sum_{i=1}^{N} p_i^2 (x_i - \mu) \geq 0 \tag{22}$$

*Proof.* By denoting $\delta_i = x_i - \mu$ and substituting it into Equation (22) together with eq. (19), we obtain

$$\sum_{i=1}^{N} p_i^2 (x_i - \mu) = \frac{1}{\left( \sum_{j=1}^{N} e^{\frac{x_j}{\tau}} \right)^2} \sum_{i=1}^{N} e^{\frac{2x_i}{\tau}} \delta_i = \frac{1}{\left( \sum_{j=1}^{N} e^{\frac{\delta_j}{\tau}} \right)^2} \sum_{i=1}^{N} e^{\frac{2\delta_i}{\tau}} \delta_i$$

Note that we can replace $x_i$ by $\delta_i$ due to the translation invariance of the softmax. Thus, it remains to show that:

$$\sum_{i=1}^{N} e^{\frac{2\delta_i}{\tau}} \delta_i \geq 0$$

By expressing the equation from Lemma A.1 in terms of $\delta_i$ we get

$$\sum_{i=1}^{N} p_i (x_i - \mu) = \frac{1}{\sum_{j=1}^{N} e^{\frac{x_j}{\tau}}} \sum_{i=1}^{N} e^{\frac{x_i}{\tau}} \delta_i = \frac{1}{\sum_{j=1}^{N} e^{\frac{\delta_j}{\tau}}} \sum_{i=1}^{N} e^{\frac{\delta_i}{\tau}} \delta_i = 0$$

Thus,

$$\sum_{i=1}^{N} e^{\frac{\delta_i}{\tau}} \delta_i = 0 \tag{23}$$

Split the sum in eq. (23) into two parts, first sum over the elements $\delta_i \geq 0$ and second over $\delta_i < 0$, as following:

$$0 = \sum_{i=1}^{N} e^{\frac{\delta_i}{\tau}} \delta_i = \sum_{\delta_i \geq 0} e^{\frac{\delta_i}{\tau}} \delta_i + \sum_{\delta_i < 0} e^{\frac{\delta_i}{\tau}} \delta_i \leq$$

$$\leq \sum_{\delta_i \geq 0} \underbrace{e^{\frac{\delta_i}{\tau}}}_{\geq 1} e^{\frac{\delta_i}{\tau}} \delta_i + \sum_{\delta_i < 0} \underbrace{e^{\frac{\delta_i}{\tau}}}_{< 1} e^{\frac{\delta_i}{\tau}} \delta_i =$$

$$= \sum_{i=1}^{N} e^{\frac{\delta_i}{\tau}} e^{\frac{\delta_i}{\tau}} \delta_i = \sum_{i=1}^{N} e^{\frac{2\delta_i}{\tau}} \delta_i = \sum_{i=1}^{N} p_i^2 (x_i - \mu)$$

$\square$

**Theorem A.3.** *The variance $\sigma_{\boldsymbol{p}}^2$ is monotonically decreasing with temperature $\tau$.*

*Proof.* By taking the derivative of the $\sigma_{\boldsymbol{x}}^2$ with respect to the $\tau$ we get.

$$
\frac{\partial \sigma_{\boldsymbol{p}}^2}{\partial \tau} = 2\frac{1}{N}\sum_{i=1}^{N}(p_i - \frac{1}{N})\frac{\partial p_i}{\partial \tau} = -2\frac{1}{N}\sum_{i=1}^{N}(p_i - \frac{1}{N})p_i(x_i - \sum_j x_j p_j)\frac{1}{\tau^2} =
$$

$$
= -\frac{1}{N}\frac{2}{\tau^2}\sum_{i=1}^{N}\left(p_i^2 x_i - \frac{1}{N}p_i x_i - p_i^2 \mu + \frac{1}{N}p_i \mu\right) =
$$

$$
= -\frac{2}{\tau^2}\frac{1}{N}\left(\sum_{i=1}^{N}p_i^2 x_i - \mu\sum_{i=1}^{N}p_i^2 - \frac{1}{N}\underbrace{\sum_{i=1}^{N}p_i(x_i - \mu)}_{0}\right) =
$$

$$
= -\frac{2}{\tau^2}\frac{1}{N}\underbrace{\sum_{i=1}^{N}p_i^2(x_i - \mu)}_{\geq 0} < 0
$$

Note that $\sum_{i=1}^{N}p_i^2(x_i - \mu) \geq 0$ as follows from the Lemma A.2. $\qquad\square$

**Lemma A.4.** *The following holds*

$$
\sum_{i=1}^{N}p_i x_i^2 - \mu^2 = \sum_{i=1}^{N}p_i(x_i - \mu)^2
$$

*Proof.*

$$
\sum_{i=1}^{N}p_i x_i^2 - \mu^2 = \sum_{i=1}^{N}p_i(x_i - \mu + \mu)^2 - \mu^2 =
$$

$$
= \sum_{i=1}^{N}p_i(x_i - \mu)^2 + 2\sum_{i=1}^{N}p_i(x_i - \mu)\mu + \mu^2 - \mu^2 =
$$

$$
= \sum_{i=1}^{N}p_i(x_i - \mu)^2 + 2\mu\underbrace{\sum_{i=1}^{N}p_i(x_i - \mu)}_{0} = \sum_{i=1}^{N}p_i(x_i - \mu)^2
$$

$\qquad\square$

### A.4.1 PROOF OF THEOREM 3.2

*Proof.* To show that the entropy in eq. (7) is monotonically increasing with temperature, we first show that the entropy of the single row eq. (20) of the SA matrix is monotonically increasing. To that end, we take a derivative $\frac{\partial H}{\partial \tau}$ of the entropy with respect to the temperature and show that is always positive.

Denote $S = \sum_{j=1}^{N}e^{\frac{x_j}{\tau}}$

The derivative of the single entry of $p$ with respect to the temperature is given by:

$$\frac{\partial p_i}{\partial \tau} = \frac{(-\frac{x_i}{\tau^2})e^{\frac{x_i}{\tau}}S - e^{\frac{x_i}{\tau}}\sum_{j=1}^{N}e^{\frac{x_j}{\tau}}(-\frac{x_j}{\tau^2})}{S^2} =$$

$$= -\frac{x_i}{\tau^2}p_i - p_i\sum_{j=1}^{N}p_j(-\frac{x_j}{\tau^2}) =$$

$$= -\frac{1}{\tau^2}p_i(x_i - \sum_{j}^{N}x_jp_j) =$$

$$= -\frac{1}{\tau^2}p_i(x_i - \mu)$$

The derivative of the entropy from eq. (20) with respect to the temperature:

$$\frac{\partial H}{\partial \tau} = -\sum_{i=1}^{N}\left(\log_2(p_i) + \frac{1}{\ln 2}\right)\frac{\partial p_i}{\partial \tau} = \sum_{i=1}^{N}\left(\log_2(p_i) + \frac{1}{\ln 2}\right)\frac{1}{\tau^2}p_i(x_i - \mu) =$$

$$= \frac{1}{\tau^2}\left(\sum_{i=1}^{N}p_ix_i\log_2(p_i) - \sum_{i=1}^{N}p_i\log_2(p_i)\mu + \frac{1}{\ln 2}\sum_{i=1}^{N}p_ix_i - \sum_{i=1}^{N}p_i\frac{1}{\ln 2}\mu\right) =$$

$$= \frac{1}{\tau^2}\left(\sum_{i=1}^{N}p_ix_i\log_2(p_i) - \mu\sum_{i=1}^{N}p_i\log_2(p_i)\right) =$$

$$= \frac{1}{\tau^2}\sum_{i=1}^{N}p_i\log_2(p_i)(x_i - \mu) =$$

$$= \frac{1}{\tau^2}\sum_{i=1}^{N}p_i\left(\frac{x_i}{\tau} - \log_2 S\right)(x_i - \mu) =$$

$$= \frac{1}{\tau^2}\left(\frac{1}{\tau}\sum_{i=1}^{N}p_ix_i^2 - \log_2 S\sum_{i=1}^{N}p_ix_i + \log_2 S\mu - \frac{1}{\tau}\mu\sum_{i=1}^{N}p_ix_i\right) =$$

$$= \frac{1}{\tau^3}\left(\sum_{i=1}^{N}p_ix_i^2 - \mu^2\right) \underset{*}{=} \frac{1}{\tau^3}\sum_{i=1}^{N}p_i(x_i - \mu)^2 > 0$$

Where, $*$ follows from Lemma A.4.

Finally, since the entropy in eq. (6) is the average entropy of the rows, the Theorem 3.2 follows.

$$H(\boldsymbol{P}^{(\text{SM})}) = \frac{1}{N}\sum_{i=1}^{N}H(\boldsymbol{p}_i) \tag{24}$$

$\square$

### A.4.2 Proof of Theorem 3.3

The proof is based on the Principal Component Analysis (PCA) (Jolliffe, 2011) of the stochastic matrix $\boldsymbol{P}$.

*Proof.* To start our proof, we first need to center matrix $\boldsymbol{P}$ in eq. (6) by deflating the first eigenvalue. Let $\boldsymbol{\mu} = \frac{1}{N}\boldsymbol{P}^{\top}\mathbf{1} = \frac{1}{N}\sum_{i=1}^{N}\boldsymbol{P}_{ij}$ the vector mean of $\boldsymbol{P}$. Note that $\boldsymbol{v}_1 = \mathbf{1}$ is the eigenvector of $\boldsymbol{P}$ corresponding to the first eigenvalue $\lambda_1 = 1$ for which the following holds:

$$\boldsymbol{\mu}^{\top}\boldsymbol{v}_1 = \frac{1}{N}\mathbf{1}^{\top}\boldsymbol{P}\mathbf{1} = \frac{1}{N}\sum_{i=1}^{N}\sum_{j=1}^{N}\boldsymbol{P}_{ij} = 1 \tag{25}$$

Thus, according to the Wielandt deflation theorem the eigenvalues of the matrix $\overline{\boldsymbol{P}} = \boldsymbol{P} - \lambda_1 \boldsymbol{v}_1 \boldsymbol{\mu}^\top$ are $0, \lambda_2, \ldots, \lambda_N$.

Furthermore, it is important to note that the matrix $\overline{\boldsymbol{P}}$ is centered in both rows and columns as follows:

$$\sum_i^N \overline{\boldsymbol{P}}_{ij} = \mathbf{1}^\top \overline{\boldsymbol{P}} = \mathbf{1}^\top \boldsymbol{P} - \lambda_1 \mathbf{1}^\top \boldsymbol{v}_1 \boldsymbol{\mu}^\top = \mathbf{1}^\top \boldsymbol{P} - N \boldsymbol{\mu}^\top = \mathbf{1}^\top \boldsymbol{P} - \mathbf{1}^\top \boldsymbol{P} = 0$$

$$\sum_j^N \overline{\boldsymbol{P}}_{ij} = \overline{\boldsymbol{P}} \mathbf{1} = \boldsymbol{P} \mathbf{1} - \lambda_1 \boldsymbol{v}_1 \boldsymbol{\mu}^\top \mathbf{1} = \mathbf{1} - \boldsymbol{v}_1 \frac{1}{N} \mathbf{1}^\top \boldsymbol{P} \mathbf{1} = \mathbf{1} - \mathbf{1} = 0$$

Therefore, the empirical covariance matrix of $\boldsymbol{P}$ can be expressed as:

$$\boldsymbol{\Sigma_P} = \left( \boldsymbol{P} - \lambda_1 \boldsymbol{v}_1 \boldsymbol{\mu}^\top \right)^\top \left( \boldsymbol{P} - \lambda_1 \boldsymbol{v}_1 \boldsymbol{\mu}^\top \right) = \overline{\boldsymbol{P}}^\top \overline{\boldsymbol{P}}$$

Denote by $\bar{\boldsymbol{v}}_1, \ldots, \bar{\boldsymbol{v}}_N$ the eigenvectors of $\overline{\boldsymbol{P}}$, where $\bar{\boldsymbol{v}}_1$ corresponds to the eigenvalue $\lambda_1 = 0$.

The variance in $\boldsymbol{P}$ along the direction of the principal component $\bar{\boldsymbol{v}}_i$ can be expressed as:

$$\sigma_{\bar{\boldsymbol{v}}_i}^2 = \frac{\bar{\boldsymbol{v}}_i^* \boldsymbol{\Sigma_P} \bar{\boldsymbol{v}}_i}{\bar{\boldsymbol{v}}_i^* \bar{\boldsymbol{v}}_i} = \bar{\boldsymbol{v}}_i^* \overline{\boldsymbol{P}}^\top \overline{\boldsymbol{P}} \bar{\boldsymbol{v}}_i = \left\| \overline{\boldsymbol{P}} \bar{\boldsymbol{v}}_i \right\|^2 = \lambda_i^2 \| \bar{\boldsymbol{v}}_i \|^2 = \lambda_i^2$$

Therefore, the variance is maximized when $\bar{\boldsymbol{v}}_i = \bar{\boldsymbol{v}}_2$, and $\lambda_2^2 = \sigma_{\bar{\boldsymbol{v}}_2}^2$ represents the amount of variance in the direction specified by the largest principal component of $\overline{\boldsymbol{P}}$. $\qquad\square$

Thus, $\lambda_2$ indicates the level of variability of matrix $\boldsymbol{P}$ along the direction specified by the major principal component of $\overline{\boldsymbol{P}}$. Moreover, according to Theorem A.3, if the rank of matrix $\boldsymbol{P}$ remains unchanged, the variability should decrease as the temperature increases. However, if the matrix $\boldsymbol{P}$ is biased towards a particular column, the variability within the columns decreases, resulting in a smaller value of $\lambda_2$. Consequently, we can infer that the spectral gap, defined as $\gamma = 1 - |\lambda_2| = 1 - \sigma_{\bar{\boldsymbol{v}}_2} \leq 1$, increases with the temperature only when the stochastic matrix is unbiased.

## A.5 PROOF OF PROPOSITION 3.1

*Proof.* Generally, the product of two independent Gaussian variables has a density in the form of a modified Bessel function of the second kind (Weisstein, 2003). When the vector dimensions are sufficiently large, the Central Limit Theorem implies that the distribution of the dot product between $\boldsymbol{q}$ and $\boldsymbol{k}$ can be approximated by a Gaussian distribution with zero mean and variance $\sigma^2$. As mentioned in section 3, the variance of $\boldsymbol{q}^\top \boldsymbol{k}$ can be expressed as $\sigma^2 = \sigma_q^2 \sigma_k^2 + C_{\text{cross}}$, where $C_{\text{cross}} = \text{Cov}(\boldsymbol{q}^2, \boldsymbol{k}^2) - \text{Cov}(\boldsymbol{q}, \boldsymbol{k})^2$ is the cross-covariance of the squared queries and keys (Goodman, 1960).

Therefore, due to the exponent in eq. (6), the numerator is approximately a log-normal variable with zero mean and variance $\sigma^2$. To address the denominator, we must consider a sum of log-normal variables. Fortunately, Fenton (1960) theorem states that for moderate values of $\sigma^2$, the sum of zero-mean i.i.d. log-normal variables can be approximated by a log-normal variable with variance $\sigma_\Sigma^2$ and mean $\mu_\Sigma$ where:

$$\sigma_\Sigma^2 = \ln \left( \frac{1}{N} \left( e^{\sigma^2} - 1 \right) + 1 \right); \quad \mu_\Sigma = \ln N + (\sigma^2 - \sigma_\Sigma^2)/2$$

For large $N$ and moderate values of $\sigma^2$, we have $\sigma_\Sigma^2 \ll \sigma^2$, and we can omit the $\sigma_\Sigma^2$ term for simplicity. Finally, since the ratio of log-normal variables remains log-normal with mean $-\mu_\Sigma$ and variance $\sigma^2$, Proposition 3.1 follows. $\qquad\square$

To empirically validate the assumption made in Proposition 3.1, we measure the variance and mean of the SA and compare them to the values predicted by the theory. The results are presented in Figure 5a, which shows that the measured statistics closely match the theoretical predictions.

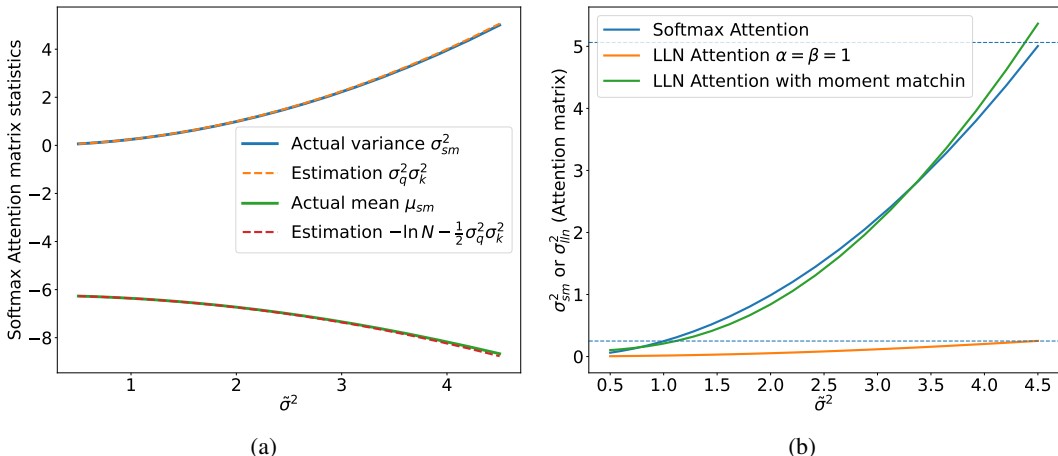

(a)                      (b)

Figure 5: (a) The variance and mean of the SA matrix with respect to the input variance. Measurements perfectly match theoretical estimation. (b) The variance of the SA and LLN Attention before and after performing the moment matching procedure.

### A.6    PROOF OF PROPOSITION 4.1

*Proof.* To demonstrate that the LLN attention matrix approximately follows a log-normal distribution, we examine a single entry of the attention matrix, which is a dot product of two vectors $q$ and $k$. The nominator $\langle e^{\alpha q}, e^{\beta k} \rangle = \sum_i^d e^{\alpha q_i + \beta k_i}$ represents a sum of $d$ log-normal variables with zero mean and variance $\tilde{\sigma}^2$, where

$$\tilde{\sigma}^2 = \alpha^2 \sigma_q^2 + \beta^2 \sigma_k^2 \tag{26}$$

Similarly, the denominator $\sum_{j=1}^N \langle e^{\alpha q}, e^{\beta k_j} \rangle$ of the LLN Attention matrix, which is also a sum of log-normal variables. According to (Fenton, 1960), for moderate values of $\tilde{\sigma}^2$, the distribution of the sum of log-normal variables can be approximated by another log-normal distribution at the right tail. Since the ratio of the log-normal variables is also log-normal, we can approximate the distribution of the LLN Attention matrix by a log-normal distribution.

To determine the relationship between the variance of the LLN Attention matrix $\sigma_{LLN}^2$ and variances of queries and keys $\sigma_q^2, \sigma_k^2$, we need to estimate the variance of a sum of log-normal variables. Following the approach of (Romeo et al., 2003), we will divide our analysis into three cases: narrow $\tilde{\sigma}^2 \ll 1$, moderate $\tilde{\sigma}^2 \lesssim 1$ and broad $\tilde{\sigma}^2 > 1$.

Denote variance of the sum in nominator by $\sigma_{nom}^2$ and variance of the denominator by $\sigma_{den}^2$.

#### NARROW CASE

If $0 < \sigma_q^2, \sigma_k^2 \ll 1$ then the values are small such that even close to zero, thus we can approximate $e^{\alpha q_i} \approx 1 + \alpha q_i$ and $e^{\beta k_i} \approx 1 + \beta k_i$, thus:

$$\sigma_{nom}^2 \approx d(\alpha^2 \sigma_q^2 + \beta^2 \sigma_k^2) = d\tilde{\sigma}^2; \quad \sigma_{den}^2 \approx N(\alpha^2 \sigma_q^2 + \beta^2 \sigma_k^2) = N\tilde{\sigma}^2 \tag{27}$$

#### MODERATE CASE

When $\tilde{\sigma}^2$ is relatively small (i.e., $\lesssim 1$), we can use (Fenton, 1960) method. The approximation is given by:

$$\sigma_{nom}^2 \approx \ln\left[\frac{\left(e^{\alpha^2 \sigma_q^2 + \beta^2 \sigma_k^2} - 1\right)}{d} + 1\right] = \ln\left[\frac{\left(e^{\tilde{\sigma}^2} - 1\right)}{d} + 1\right] \tag{28}$$

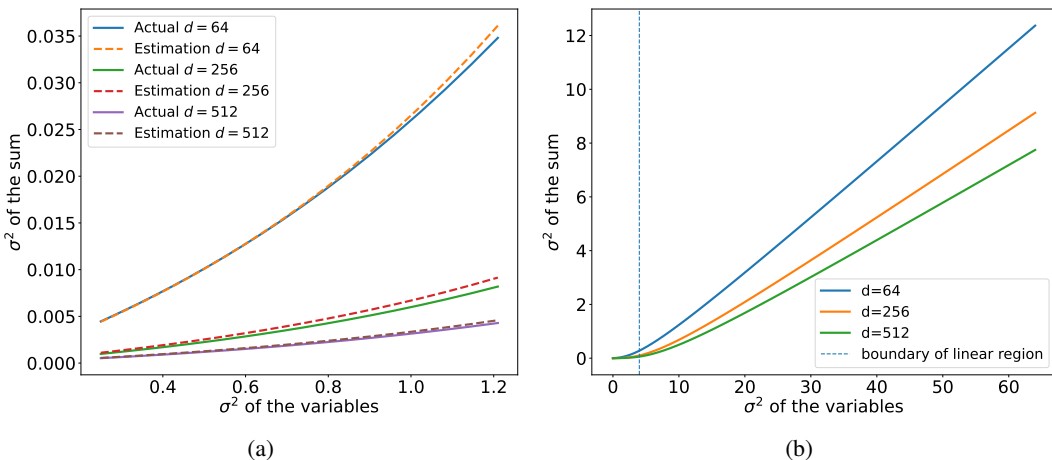

(a)                                    (b)

Figure 6: The variance of the sum of $d$ log-normally distributed inputs, each with a variance of $\sigma^2$ and a zero mean, is effectively estimated by the (Fenton, 1960) method. (a) In moderate scenario where $\sigma^2$ lies within the range of $[0.2, 1.2]$, theoretical estimations (depicted as dashed lines) aligned with empirical results. (b) In the broad case, for sufficiently large $\sigma^2$ the variance of the log-normal sum grows linearly with the variance of the input variables.

Similarly, for the denominator:

$$\sigma^2_{den} \approx \ln \left[ \frac{\left( e^{\tilde{\sigma}^2} - 1 \right)}{N} + 1 \right] \tag{29}$$

In fig. 6a, we demonstrate the empirical evaluation of the correctness of Fenton approximation when $\sigma^2 \in [0.2, 1.2]$.

BROAD CASE

In the broad case, when $\tilde{\sigma}^2$ is large (i.e., $\tilde{\sigma}^2 \gg 1$), it is not possible to find a closed-form approximation for the log-normal sum. However, we can observe that the sum of exponents is dominated by the largest term, which corresponds to the maximum value. This maximum value grows linearly with the spread of queries and keys under the Gaussian assumption. Consequently, according to (Romeo et al., 2003), the resulting variance $\tilde{\sigma}^2$ is also linearly proportional to $\tilde{\sigma}^2$ with some constants $a_1, a_2$ and $b_1, b_2$:

$$\sigma^2_{nom} \approx a_1(\alpha^2 \sigma_q^2 + \beta^2 \sigma_k^2) + b_1 = a_1 \tilde{\sigma}^2 + b_1 \tag{30}$$

$$\sigma^2_{den} \approx a_2(\alpha^2 \sigma_q^2 + \beta^2 \sigma_k^2) + b_2 = a_2 \tilde{\sigma}^2 + b_2 \tag{31}$$

We empirically evaluated the linear dependency assumption of the sum of log-normally distributed inputs and showed its validity in fig. 6b.

Finally, the variance of the LLN Attention matrix is a sum of the nominator and denominator variances, i.e.:

$$\sigma^2_{\text{lln}} = \sigma^2_{nom} + \sigma^2_{den} \tag{32}$$

By denoting $a = a_1 + a_2$ and $b = b_1 + b_2$ for a broad case we get:

$$\sigma^2_{\text{lln}} = a\tilde{\sigma}^2 + b = a(\alpha^2 \sigma_q^2 + \beta^2 \sigma_k^2) + b \tag{33}$$

$\square$

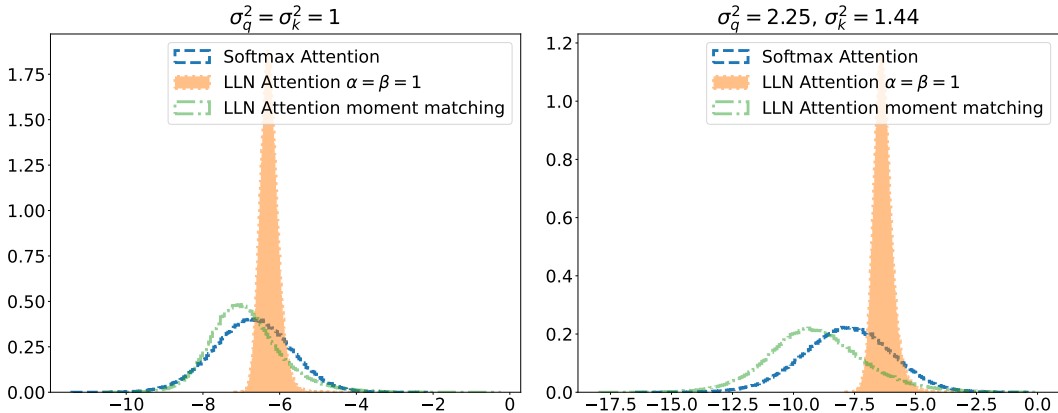

Figure 7: Histogram of the SA and LLN Attention before and after performing the moment matching.

### A.7 MOMENT MATCHING

As shown in Figure 5b, we are interested in handling a broad range of $\tilde{\sigma}^2$ values, particularly those in $[1, 4]$. Using Proposition 4.1, we can find the constants $a$ and $b$ that satisfy the requirement $\sigma_{lln}^2 = \sigma_{sm}^2$ of the broad case through moment matching. To do this, we perform linear interpolation between the variances of the LLN and Softmax Attention by injecting uncorrelated Gaussian inputs to both attentions and measuring their output variances according to the following equation:

$$\sigma_{sm}^2 = \sigma_{lln}^2 = a\tilde{\sigma}^2 + b \tag{34}$$

Once we have determined the values for $a$ and $b$, we can substitute them into Equation (10) to obtain the optimal values for $\alpha$ and $\beta$.

As depicted in Figure 5b, the variance of the LLN Attention without moment matching (i.e., $\alpha = \beta = 1$) is much smaller than that of the Softmax Attention and exhibits a nearly linear trend. However, the variance of the LLN Attention with moment matching approximates that of the Softmax Attention. Additionally, the histogram shown in Figure 7 suggests that the LLN Attention distribution closely follows that of the Softmax Attention. Despite that, the two distributions have slightly different means because we only match their variances.

### A.8 EXPERIMENTS

In this section, we present more experimental results and ablation studies of our method.

#### A.8.1 PRE-TRAINING OF ROBERTA MODEL

We train the bidirectional RoBERTa-base encoder model (Liu et al., 2019) using LLN Attention on the WikiText-103 corpus (Merity et al., 2018). During pre-training, we monitor the convergence of the model and compare its performance to the SA model. In Figure 8a we show the training and validation loss of the RoBERTa-base model during pre-training with LLN Attention, as well as its comparison to SA. The loss curve of LLN Attention closely follows the SA, indicating similar convergence behavior. We used the Fairseq framework (Ott et al., 2019) for all experiments with the default configuration and hyperparameters of the RoBERTa-base model.[4]

We perform the training with FP16 precision, which can cause instability during training. To test the stability of the training, we also log the inverse loss scale parameter Figure 8b. Spikes in the plot indicate a decrease in the loss scale due to large gradients. As can be seen from the figure, the maximum inverse scale of LLN Attention does not exceed that of the SA, which is important to ensure similar stability during training as with SA.

---

[4]https://github.com/facebookresearch/fairseq/blob/main/examples/roberta/README.md

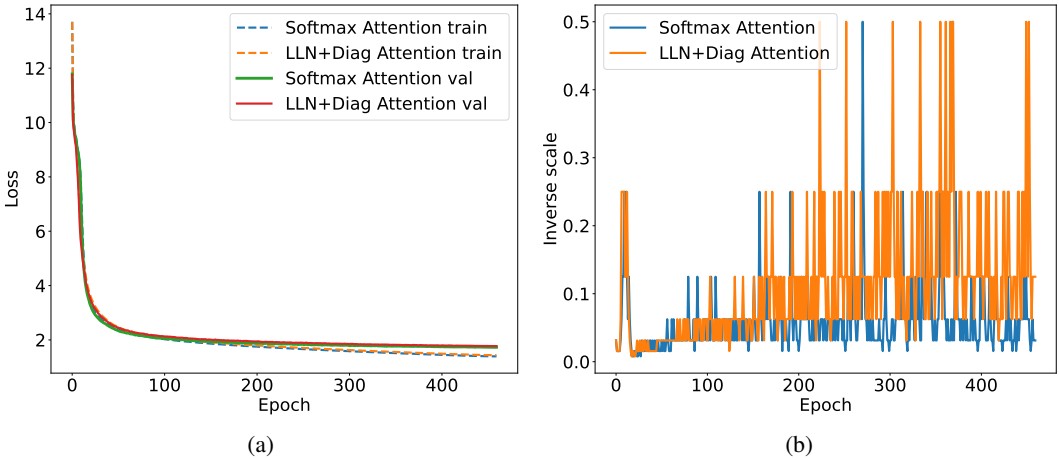

Figure 8: (a) Training and validation loss comparison of RoBERTa-base model pre-training using LLN Attention and SA. (b) Inverse of the loss scale during training of RoBERTa-base model.

### A.8.2  IMAGE CLASSIFICATION

To test LLN Attention on Vision task, we evaluate it on the Vision Transformer model using vit-pytorch [5] code base. Our ViT model consists of twelve layers and 128 embedding sizes. We train this model for 100 epochs on Dogs vs Cats dataset [6] with LLN and Softmax Attention. The results in Table 3 show that LLN Attention performs on par with SA while outperforming the Linformer (Wang et al., 2020) method.

| Softmax | LLN+Diag | Linformer |
|---------|----------|-----------|
| 81.37   | 81.72    | 79.89     |

Table 3: Accuracy [%] of the ViT model trained on Dogs vs Cats dataset with Softmax, LLN (ours) and Linformer (Wang et al., 2020) Attention.

### A.8.3  LONG RANGE ARENA

| method | Time[s] | | | | | Memory[Mb] | | | | |
|--------|-----|------|-------|------|-------|-------|------|------|------|------|
|        | TC  | LO   | RE    | PF   | IC    | TC    | LO   | RE   | PF   | IC   |
| Softmax | 21468 | 5905 | 21866 | 6754 | 13228 | 17108 | 4458 | 8934 | 4817 | 9632 |
| Reformer | 4610 | 2439 | 4714 | 4694 | 8737 | 3261 | 1631 | 3008 | 3258 | 6514 |
| Performer | 3456 | 1966 | 3761 | 3553 | 13169 | 2176 | 1122 | 2178 | 2180 | 4353 |
| Skyformer | 4523 | 2970 | 5602 | 5240 | 9347 | 3068 | 1697 | 2974 | 4041 | 8079 |
| LLN + Diag | 3043 | 1774 | 3135 | 3042 | 4053 | 1641 | 821 | 1586 | 1639 | 3276 |

Table 4: Comparison of memory[Mb] and running time [s] of LLN Attention with Reformer(Kitaev et al., 2020), Performer(Choromanski et al., 2020) and Skyformer(Chen et al., 2021) linear attention methods and SA baseline.

We use the Long Range Arena (LRA) (Tay et al., 2020c) benchmark to evaluate LLN Attention on longer sequences. LRA benchmark requires a sequence length between 1k and 4k, depending on the task. To that end, we used a code base provided by Skyformer (Chen et al., 2021) [7]. We compare the LRA score in addition to the memory and computation complexity of LLN Attention with Reformer(Kitaev et al., 2020), Performer(Choromanski et al., 2020), and Skyformer(Chen et al., 2021) linear methods as well as regular SA. According to the Table 4, LLN Attention requires much less

---

[5]https://github.com/lucidrains/vit-pytorch

[6]https://www.kaggle.com/competitions/dogs-vs-cats-redux-kernels-edition/data

[7]https://github.com/pkuzengqi/Skyformer

memory and time compared to other methods while achieving a similar average LRA score as SA Table 5.

| method | Text (4k) | ListOps (2k) | Retrieval (4k) | Pathfinder (1k) | Image (1k) | AVG |
|--------|-----------|--------------|----------------|-----------------|------------|-----|
| Softmax | 60.41 | 38.05 | 79.95 | 71.3 | 37.2 | 57.38 |
| Reformer | 61.27 | 37.05 | 78.74 | 67.23 | 44.04 | 57.67 |
| Performer | 57.85 | 37.8 | 80.5 | 62.58 | 37.56 | 55.26 |
| Skyformer | 60.88 | 39.36 | 81.54 | 70.23 | 32.64 | 56.93 |
| LLN + Diag | 60.72 | 38.91 | 81.21 | 69.81 | 38.65 | **57.86** |

Table 5: LRA score of LLN Attention with Reformer(Kitaev et al., 2020), Performer(Choromanski et al., 2020) and Skyformer(Chen et al., 2021) linear attention methods and SA baseline.

### A.8.4 LLN ATTENTION CONCENTRATION - ABLATION STUDY

In this section, we analyze the impact of the LLN Attention temperature of the Vision Transformer (ViT) model trained on the Dogs vs Cats dataset. First, we record the values of $\alpha$ and $\beta$ produced by the moment matching procedure during training. According to Figure 9, the values of $\alpha$ and $\beta$ obtained during moment matching lay within the range of $(2; 2.2)$. Furthermore, since the temperature of the LLN Attention, as defined in Equation (11), decreases as the hyper-parameters $\alpha$ and $\beta$ increase. To assess the influence of the temperature, we train the model with various fixed values of hyper-parameters $\alpha$ and $\beta$ and record the resulting accuracy. In Figure 10a, we see that when $\alpha$ and $\beta$ values are smaller than the moment matching range, i.e., less than 2, the LLN Attention concentration is insufficient due to the high temperature, leading to accuracy degradation. Conversely, when $\alpha$ and $\beta$ values lay within the range of moment matching values or larger ($\alpha, \beta \geq 2$), the concentration is sufficient for the model to achieve the desired accuracy.

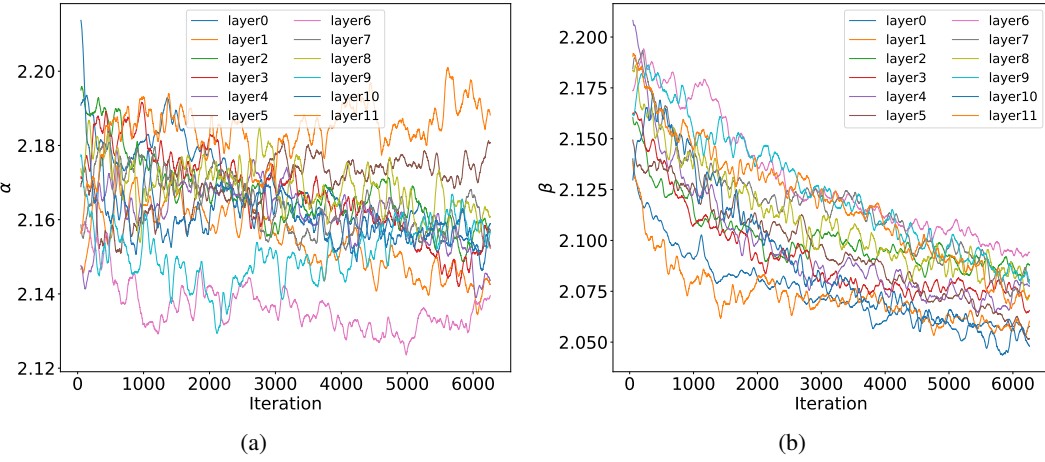

Figure 9: Change in the $\alpha$ parameter (a) and $\beta$ (b) during training of ViT model on Dogs vs Cats dataset.

We highlight the risks associated with surpassing the moment matching range by increasing $\alpha$ and $\beta$. In particular, larger values of these parameters may risk the stability of the training process due to increased gradients, a concern that becomes especially noticeable when training models in Float16 format. The risk of utilizing the Float16 data type stems from the reduced precision, smaller dynamic range, risks of gradient overflow, and the requirement to maintain the loss scaling. Moreover, the lower precision of Float16 may result in information loss during computations and numerical instability.

Accordingly, exceeding the moment matching values of $\alpha$ and $\beta$ is practically undesirable, particularly in the context of training with Float16. In Figure 10b, we illustrate this phenomenon by presenting the loss scale during the training of the deit-tiny model(Touvron et al., 2020) in the Float16 format. We see that for large values of $\alpha = \beta = 4$, the inverse of the loss scale is significantly larger,

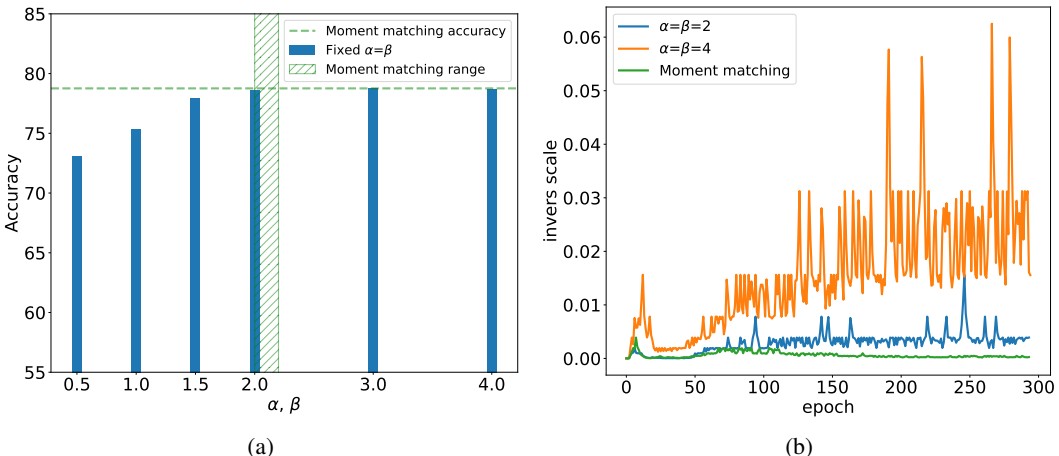

Figure 10: (a) Accuracy of ViT trained on Dogs vs Cats dataset with LLN Attention and different values of $\alpha$ and $\beta$. (b) Inverse of the loss scale during training of deit-tiny model for different fixed values of $\alpha$ and $\beta$ as well as for moment matching.

compared to $\alpha = \beta = 2$, indicating increased gradients and the potential of training instability or even failure. Therefore, to achieve the desired accuracy and allow stable training, it is crucial to maintain the temperature in the "sweet spot" specified by the moment matching values.

