# OpenReview forum: "Linear Log-Normal Attention with Unbiased Concentration"
_ICLR.cc/2024/Conference — ICLR 2024 poster_

### Official Review · Reviewer_aZn4 · 2023-10-30

**Soundness:** 3 good
**Presentation:** 3 good
**Contribution:** 3 good
**Rating:** 6
**Confidence:** 3

**Summary:**

This study scrutinizes the self-attention mechanism, focusing on the distribution and concentration behavior inherent in the original self-attention model, and introduces an innovative linear log-normal attention mechanism. Empirical evidence gathered from widely recognized natural language processing benchmarks demonstrates that the proposed method surpasses other linearized attention alternatives in performance.

**Strengths:**

1. Linearized Attention has emerged as a crucial subject in contemporary transformer architectures.

2. Empirical evidence demonstrates its efficacy.

**Weaknesses:**

1. The experimental results across various Natural Language Processing (NLP) tasks do not exhibit significant improvements.

2. The absence of latency comparison is a drawback; providing only theoretical analysis is insufficient without practical, comparative data.

**Questions:**

See weakness

---

> ### Author Response · Authors · 2023-11-15
> **Response to the Reviewer aZn4**
>
> We would like to thank the reviewer for their time and feedback. We hope our response addresses the reviewers' concerns.
>
> > The experimental results across various Natural Language Processing (NLP) tasks do not exhibit significant improvements.
> - Please note that Nystromformer has a different baseline score, which is higher than our baseline (as mentioned in Table 1). Considering accuracy degradation, Nystromformer has a drop of ~1.5\% of its baseline in MNLI and QNLI (see Table 2 in Nystromformer paper), whereas LLN Attention has a drop of ~0.5\% in these datasets. We have improved Table 1 to clarify this point. Beyond these, LLN Attention outperforms all other methods achieving results very close to the baseline (a drop of 0.1\% on average). Taking all the above into account, and since the score of Linear Attention is limited by the baseline score, we disagree with the reviewer regarding the insufficient improvement as achieving the baseline should be sufficient for the method's success. Moreover, we would like to emphasize that the novelty of our method is mostly in its theoretical foundation and analysis and, hence, has a potential for other, more complicated tasks.
>
>     Still, we agree with the reviewer that our empirical evaluation requires improvement. We extended our experimental results with the Image Classification task and LRA which we added in Appendix A.8.2 and A.8.3 in the revised paper. We also added those results above (response to reviewer zqGD).
>
> > The absence of latency comparison is a drawback; providing only theoretical analysis is insufficient without practical, comparative data.
>
> - Regarding latency comparison, we would like to point the reviewer's attention to Table 2, where we present the memory and speed of LLN Attention. The results in Table 2 confirm that LLN Attention scales linearly with sequence length. Moreover, it can handle at least four times longer sequences than SA on the GPU with 32Gb RAM. While the memory of both Nystromformer and LLN Attention is approximately the same (both allocate similar size activations for backward computation), the LLN Attention demonstrates better latency than the Nystromeformer.
>
>     Also, please note that we added latency and memory measurements on the Long Range Arena (LRA) benchmark in Appendix A.8.3 of the revised paper. According to the results in A.8.3, LLN Attention requires much less memory and time compared to other methods while achieving a similar average LRA score as SA.

---

> > ### Comment · Reviewer_aZn4 · 2023-11-22
> > **Change my score based on Author's rebuttal**
> >
> > After reading author's rebuttal, i decide to upgrade my score.

---

### Official Review · Reviewer_m7qC · 2023-10-30

**Soundness:** 3 good
**Presentation:** 3 good
**Contribution:** 3 good
**Rating:** 5
**Confidence:** 4

**Summary:**

This paper proposed a linear log-normal attention with unbiased concentration to deal with the quadratic time and memory complexity of self-attention. Experiments on several NLP tasks show its effectiveness.

**Strengths:**

The analysis of this this paper is good and the design of linearized attention is interesting. The experiments and the visualized results are easy to follow.

**Weaknesses:**

1. The main concern of this paper is its results. Compared with Nystromformer, which published two years ago, this paper dose not improve enough, from both results and efficiency, as shown in  Table 1 and Table 2.
2. Only NLP tasks are adopted, but computer vision based tasks are neglected.
3. Some related works are missing, such as KVT (ECCV2022).
4. The experiments are not extensive.

**Questions:**

See weakness.

---

> ### Author Response · Authors · 2023-11-19
> **Response to the Reviewer m7qC**
>
> We thank the reviewer for dedicating their time to assess our work and providing constructive feedback. We also value the reviewer's recognition of our theoretical analysis and evaluation efforts and we tried to address all the points raised by the reviewer.
>
> > The main concern of this paper is its results. Compared with Nystromformer, which published two years ago, this paper dose not improve enough, from both results and efficiency, as shown in Table 1 and Table 2.
> -   Please note that Nystromformer has a different baseline score, which is higher than our baseline (as mentioned in Table 1). Considering accuracy degradation, Nystromformer has a drop of ~1.5\% of its baseline in MNLI and QNLI (see Table 2 in Nystromformer paper), whereas LLN Attention has a drop of ~0.5\% in these datasets. We have improved Table 1 to clarify this point. Furthermore, since LLN Attention achieved results very close to the baseline, it is impossible for one method to significantly outperform another. Therefore, achieving the baseline should be sufficient for the method's success.
>
>     Additionally, despite Nystromformer and LLN Attention performing similar operations and allocating comparable resources, LLN Attention demonstrates slightly faster performance than Nystromformer, as outlined in Table 2. In addition, please note that we extended our evaluation with additional experiments on the LRA benchmark, which outlines the significant advantage of LLN Attention in latency and memory over other methods (Appendix A.8.3 in the revised version of the paper). Having said that, we want to emphasize that while efficiency improvement is desirable, the primary objective of Linearized Attention is to provide asymptotically linear memory and computation complexity while preserving baseline accuracy rather than improving the latency over other methods by a small portion.
>
> > Only NLP tasks are adopted, but computer vision based tasks are neglected.
> - To address the reviewer's concern regarding the absence of the Vision tasks, we added an evaluation of the LLN Attention on Image Classification task. To that end, we trained the 12-layer ViT model on the Dogs vs Cats dataset. From the results in the Table below, LLN Attention performs on par with the SA baseline while outperforming the Linformer method. We have added those results in Appendix A.8.2 of the revised paper.
>
>     | Softmax | LLN+Diag | Linformer |
>     |:-------:|:--------:|:---------:|
>     |  81.37  |   81.72  |   79.89   |
>
> > Some related works are missing, such as KVT (ECCV2022).
> - We thank the reviewer for pointing us to the KVT work, which we previously overlooked. Indeed, the KVT method, which selects top-k elements of the attention matrix in each row, is an interesting direction. The main claim of the authors is faster convergence and higher accuracy due to noise reduction. However, it does not allow linear memory and computational complexity as it still requires computing and storing full attention matrix for backward path computations unless one implements a fused sparse kernel for KVT Attention. We thank the reviewer for this reference. We added a citation in the related work section of the revised submission.
>
> > The experiments are not extensive.
> - To address the reviewer's concern, we conducted additional experiments on the Long Range Arena (LRA) benchmark. Since those results require a lot of space, we are not copying them here. Instead, please refer to the response above (reviewer zqGD) or/and Appendix A.8.3 in the revised version of the paper.
>
> We hope those results, in addition to the Image Classification experiment, will be sufficient for the reviewer to reconsider his evaluation.

---

### Official Review · Reviewer_zqGD · 2023-11-01

**Soundness:** 2 fair
**Presentation:** 2 fair
**Contribution:** 2 fair
**Rating:** 5
**Confidence:** 4

**Summary:**

The paper proposes a new self-attention mechanism called Linear Log-Normal Attention (LLN Attention) that reduces the quadratic computational complexity of standard softmax attention while aiming to maintain comparable performance.
The authors first provide background on transformer models and the standard scaled dot-product softmax attention, as well as prior work on efficient linear attention mechanisms.
They then conduct an in-depth analysis of softmax attention, characterizing its statistical, informational, and algebraic properties.
In particular, they model the distribution of the attention matrix and prove its log-normal nature.
Furthermore, they analyze the concentration behavior of softmax attention using entropy and spectral gap metrics.
Based on this analysis, the authors design the LLN Attention method to emulate the log-normal distribution and concentration properties of standard softmax attention.
LLN Attention uses exponential feature embeddings of the queries and keys to induce a log-normal distribution.
The authors also introduce a temperature parameter and perform moment matching between LLN Attention and softmax attention to align their concentration behavior.
This allows LLN Attention to achieve comparable performance to softmax attention.
The paper provides proofs that LLN Attention follows a log-normal distribution and that its entropy and spectral gap vary similarly to softmax attention with respect to temperature.
Experimental results on natural language processing benchmarks demonstrate that LLN Attention outperforms other linear attention methods and achieves competitive accuracy compared to softmax attention.

**Strengths:**

- This paper made contributions in 1) in-depth analysis and modeling of distributional and concentration properties of softmax attention; 2) Design of LLN Attention method that emulates softmax attention based on this analysis; 3) Introduction of moment matching technique to align concentration behavior; 4) Linear complexity in sequence length while maintaining softmax attention performance. The proposed LLN Attention offers an interesting and somewhat promising approach to enhance transformer scalability for long sequences.

- The paper provides a novel perspective on analyzing self-attention through statistical and information-theoretic lenses. The design of LLN Attention based on emulating properties of softmax attention is a useful technique for developing efficient linear attention. Matching the log-normal distribution and concentration patterns is a creative way to achieve comparable performance.
- The analysis provides useful insights into the workings of the widely used softmax attention. Characterizing the distribution and concentration can guide better attention design. LLN Attention demonstrates the viability of achieving softmax performance with linear complexity. This can expand the applicability of transformers to longer sequences.
- The proposed techniques like moment matching may inspire novel ways to analyze and improve attention mechanisms in future work.

**Weaknesses:**

- The theoretical analysis relies on several approximations, such as using the Fenton theorem to model log-normal sums. While justified, evaluating the accuracy of these approximations on empirical data could be beneficial.
- The paper focuses exclusively on natural language tasks. Assessing the effectiveness of LLN Attention on other modalities like computer vision with ViTs could provide more insight into its general applicability.
- Only accuracy results are reported. Including other metrics like training time, stability, convergence rate, etc. could allow more comprehensive comparison to softmax attention.
- The moment matching technique matches up to 2nd order statistics of LLN and softmax attention. Have the authors considered or tried extending to higher order statistics?
- Combining LLN Attention with sliding window or dilated patterns could be an interesting extension to handle local contexts more effectively. What do the authors think?
- The impact of different feature embedding functions besides the exponential function could be analyzed. Do they provide similar distributions and concentration?
- More ablation studies on the temperature parameter, block size in diagonal attention, layer depth, etc. could shed light on how to best configure LLN Attention.
- The quality and diversity of the tasks used for evaluation could be expanded. For instance, adding tasks with longer input sequences could better highlight benefits of LLN Attention.

**Questions:**

Please see above.

---

> ### Author Response · Authors · 2023-11-15
> **Response to the Reviewer zqGD (Part I)**
>
> We thank the reviewer for their time and comprehensive feedback, which has been invaluable in refining our work. We are working to address all the points raised by the reviewer, including more experimental results and additional ablation studies.
>
> > The theoretical analysis relies on several approximations, such as using the Fenton theorem to model log-normal sums. While justified, evaluating the accuracy of these approximations on empirical data could be beneficial.
> - As suggested by the reviewer, we added empirical justification for the Fenton approximation in Fig. 6(a) in the appendix. In addition, we included empirical justification for the broad case (Romeo et. al) in Fig. 6(b) in the appendix. We uploaded those results with the revised version of the paper. Beyond those, we would like to point reviewers' attention to the empirical justification of propositions 3.1 and 4.1 in figures 5(a) and 5(b) in the appendix of the original submission, which also indicates the validity of the Fenton approximation.
>
> > Only accuracy results are reported. Including other metrics like training time, stability, convergence rate, etc.
> - Please note that due to space limitation, we moved part of the empirical results to the appendix. Specifically, in the appendix Fig. 7 (original submission) or Fig. 8(a) (revised submission), we present the loss curve of the Roberta model. We can see, that training/validation loss of LLN Attention closely follows that of the SA. This figure justifies that the training time (in epochs) and convergence rate of LLN Attention are similar to the SA. Further, to address reviewers' request for the stability indicators, we added measurements of the loss scale during training Fig. 8(b) in the appendix of the revised paper. According to those results, the inverse of the loss scale for LLN Attention does not exceed that of the SA, which indicates similar stability during training.
>
> > The moment matching technique matches up to 2nd order statistics of LLN and softmax attention. Have the authors considered or tried extending to higher order statistics?
> - As we proved in propositions 3.1 and 4.1, the probability distributions of LLN and Softmax Attention closely resemble log-normal distribution. Notably, log-normally distributed random variable $X=e^{\sigma Z + \mu}$ is fully determined by parameters $\mu$ and $\sigma$, which are the first and second moments of Gaussian variable $Z$. Therefore, it is sufficient to match the moments of $Z$ up to 2nd order since higher order moments are determined by the first two. However, in cases where the log-normal distribution does not approximate well the Softmax (e.g. for very large values of $\sigma$ or large deviation of the input from the Gaussian distribution), high-order moments should be considered. Nevertheless, it could be very intricate in practice.
>
> > Combining LLN Attention with sliding window or dilated patterns could be an interesting extension to handle local contexts more effectively. What do the authors think?
> - Combining LLN Attention with sliding window or dilated Attention is definitely interesting as it may allow better handling of short-term interactions. We will explore those possibilities in future work. However, the drawback of these techniques is that they are more compute-intensive and require more complex implementation than the simple diag-attention method. From this perspective, if the combination of diag-attention with LLN Attention may achieve comparable results to the former, it would be beneficial concerning resources and simplicity.
>
> > The impact of different feature embedding functions besides the exponential function...
> - Concerning the ablation study with other feature embedding functions, we would like to point out the reviewer to Figure 2 in the paper. In this figure, we demonstrate the entropy and the spectral gap of other kernels where "Quadratic linear" and "ReLU linear" represent the Linearized Attention with Quadratic and ReLU feature embedding functions, respectively. We can see that both Quadratic and ReLU embedding functions are indifferent to the temperature, as a result, their concentration behavior significantly differs from the SA and may degrade performance. To test this assumption, we trained a couple of Transformer models with 256 and 512 embedding sizes and four layers each, with Linearized Attention consisting of Quadratic and ReLU feature embedding functions. As you can see in the table below, the perplexity of the linear kernels is much higher than that of the SA, which we relate to poor concentration.
>
>     | model\ppl  | softmax | quad lin | relu lin |
>     |------------|:-------:|:--------:|:--------:|
>     | medium (4, 512) |   5.9   |   9.87   |   10.32  |
>     | small (4, 256)  |  33.87  |    166   |   97.7   |

---

> ### Author Response · Authors · 2023-11-17
> **Response to the Reviewer zqGD (Part II)**
>
> > The paper focuses exclusively on natural language tasks. Assessing the effectiveness of LLN Attention on other modalities like computer vision with ViTs could provide more insight into its general applicability.
> - To address reviewers' concerns regarding the absence of the Vision tasks, we evaluated our method on the ViT model with 12 layers from [1] using the Dogs vs Cats dataset. As we can see from the table below, LLN Attention performs on par with SA while outperforming the Linformer method. We have added those results to the appendix A.8.2 of the revised paper, as well as more details of the training process.
>
>     | Softmax | LLN+Diag | Linformer |
>     |:-------:|:--------:|:---------:|
>     |  81.37  |   81.72  |   79.89   |
>
>     [1] [vit-pytorch](https://github.com/lucidrains/vit-pytorch)
>
> > The quality and diversity of the tasks used for evaluation could be expanded. For instance, adding tasks with longer input sequences could better highlight benefits of LLN Attention.
> - To address the reviewer's request for evaluation with longer sequences, we expanded our experiments on Long Range Arena (LRA) [1] tasks, which are popular for linear attention benchmarks. The sequence length of these tasks varies between 1k and 4k, depending on the task. Our experiments show that LLN Attention achieves the same average LRA score as SA while requiring much less memory and time than all other linear methods (table below). Since due to markdown issues it is hard to read table bellow, please refer to the appendix A.8.3 of the revised paper.
>
>     |                   |               |          |Time [s]|          |   |  |  |Memory|[Mb] | |
>     |---------------|:----------:|:-----------------:|:--------:|:-------:|:--------:|:----------:|:----:|:----:|:----:|:----:|
>     | method      |    TC     |  LO   |   RE    |  PF    |   IC     |  TC |  LO  |  RE  |  PF  |  IC  |
>     | Softmax     |  21468  | 5905 | 21866 | 6754 | 13228 | 17108 | 4458 | 8934 | 4817 | 9632 |
>     | Reformer   |   4610   | 2439 |  4714  | 4694 |  8737 | 3261 | 1631 | 3008 | 3258 | 6514 |
>     | Performer  |   3456   | 1966 |  3761  | 3553 | 13169 | 2176 | 1122 | 2178 | 2180 | 4353 |
>     | Skyformer  |   4523   | 2970 |  5602  | 5240 |  9347 | 3068 | 1697 | 2974 | 4041 | 8079 |
>     | LLN + Diag |   3043  | 1774 |  3135  | 3042 |  4053 | 1641 |  821 | 1586 | 1639 | 3276 |
>
>
>     | method     | Text (4k) | ListOps (2k) | Retrieval (4k) | Pathfinder (1k) | Image (1k) |  AVG  |
>     |------------|:---------:|:------------:|:--------------:|:---------------:|:----------:|:-----:|
>     | Softmax    |   60.41   |     38.05    |      79.95     |       71.3      |    37.2    | 57.38 |
>     | Reformer   |   61.27   |     37.05    |      78.74     |      67.23      |    44.04   | 57.67 |
>     | Performer  |   57.85   |     37.8     |      80.5      |      62.58      |    37.56   | 55.26 |
>     | Skyformer  |   60.88   |     39.36    |      81.54     |      70.23      |    32.64   | 56.93 |
>     | LLN + Diag |   60.72   |     38.91    |      81.21     |      69.81      |    38.65   | 57.86 |
>
>     [1] [Long Range Arena](https://arxiv.org/abs/2011.04006)

---

> ### Author Response · Authors · 2023-11-20
> **Response to the Reviewer zqGD (Part III)**
>
> > More ablation studies on the temperature parameter, block size in diagonal attention, layer depth, etc. could shed light on how to best configure LLN Attention.
> - To address reviewers request for more ablation studies, we added more experiments for the temperature and concentration of the LLN Attention in the Appendix A.8.4 in the revised submission. To assess the influence of the temperature, we train the ViT model with various fixed values of hyper-parameters $\alpha$ and $\beta$ and record the resulting accuracy. In Figure 10a, we see that when $\alpha$ and $\beta$ values are smaller than the moment matching range, i.e., less than $2$, the LLN Attention concentration is insufficient due to the high temperature, leading to accuracy degradation. Conversely, when $\alpha$ and $\beta$ values lay within the range of moment matching values or larger ($\alpha, \beta \geq 2$), the concentration is sufficient for the model to achieve the desired accuracy.
>
>     However, it is not desirable to increase the $\alpha$ and $\beta$ too much, as for larger values training may become unstable due to increasing gradients. In the Figure 10b we show loss scale of the model during training with Float16 data type. It can be seen from the figure, that for larger values of $\alpha,\beta$, the loss scale values are much smaller than for small $\alpha,\beta$, which indicates increased gradients and hence may cause instability during training. Therefore, to achieve the desired accuracy and allow stable training it is crucial to maintain the temperature in the "sweet spot" specified by the moment matching values. For more detail, please refer to the Appendix A.8.4 of the revised submission.
>
> We hope that we have addressed all the concerns raised by the reviewer by providing comprehensive responses and additional experimental results. If the reviewer finds our responses satisfactory, we would greatly appreciate the reconsideration of their evaluation.

---

### Official Review · Reviewer_xXRJ · 2023-11-02

**Soundness:** 3 good
**Presentation:** 3 good
**Contribution:** 3 good
**Rating:** 8
**Confidence:** 3

**Summary:**

The paper proposes a linear log-normal attention mechanism that has linear time and memory complexity while retaining key properties of the naive softmax attention (SA). The key properties under consideration are the log-normal distribution of the attention matrix and the monotonicity of the entropy as well as the spectral gap with temperature. Several experiments with different NLP tasks are performed to confirm the efficiency of the proposed method.

**Strengths:**

- The paper is very well-written with detailed theoretical justification and many useful intuitive explanations.

- The authors have characterized three important and interesting properties of the SA mechanism: (1) the distribution of the SA matrix $\mathbf{P}^{(SM)}$ can be approximated by a log-normal distribution, (2) the entropy $H(\mathbf{P}^{(SM)})$ is monotonically increasing with temperature, while (3) the variance of the attention matrix $\mathbf{P}^{(SM)}$ is decreasing with temperature. I have not checked the proofs line by line, but it seems to me that the proofs are correct.

- Based on these characterizations of the SA mechanism, the authors carefully selected suitable feature embedding functions $\phi$ in the linearized attention equation in a way that these characterizations still hold.

- Experiments with different NLP tasks confirmed the advantages of the proposed model in comparison with previous linear attention methods.

- In these experiments, the authors also experimentally show that the entropy and the spectral gap of the attention matrix are actually monotonically increasing, as proved in the theoretical parts.

Overall, I think this is a good paper.

**Weaknesses:**

I do not see any major weaknesses of the paper. There are a few minor weaknesses as follows:

- Some notions are introduced with formulas but without intuitive explanations. For example, why is $\tau_{sm}$ in Eq. (5) called the "temperature" of the SA? and why controls the level of exploration and exploitation?

- In Eq. (6), $P_{ij}^{(SM)}$ is used, while in Eq. (7), it is $\mathbf{P}_{ij}^{(SM)}."

- I am not sure where the proof of Theorem 3.4 is in the Appendix. Is it Theorem A.3?

**Questions:**

See weaknesses!

---

> ### Author Response · Authors · 2023-11-13
> **Response to the Reviewer xXRJ**
>
> We would like to thank the reviewer for their time and encouraging feedback. Hope our answers will clarify the questions of the reviewer.
>
> > Some notions are introduced with formulas but without intuitive explanations. For example, why is
>  in Eq. (5) called the "temperature" of the SA? and why controls the level of exploration and exploitation?
> - A general form of the softmax function with temperature is $Softmax(x, \tau) = \frac{e^{x/\tau}}{\sum_i e^{x_i/\tau}}$ where $\tau$ is temperature parameter. We show that the SA matrix can be expressed with respect to the $\tau_{sm}$ in eq.(6), which is exactly the form of the softmax with temperature.
> Softmax with temperature is frequently used in the context of Reinforcement Learning for action selection [1][2]. Controlling the temperature allows balancing between exploration and exploitation. High temperatures cause all actions to be nearly equiprobable (exploration), whereas low temperatures cause greedy
> action selections (exploitation). Similarly, in the attention mechanism, high-temperature results in equal probabilities for all tokens (exploration i.e. any token can be chosen). On the other hand, low temperature results in a large probability for one or few tokens, emphasizing it (exploitation of this particular token).
> Due to the reviewer's concern, we will improve the intuition part in the revised version of the submission.
>
> > In Eq. (6), $P_{ij}^{(SM)}$ is used, while in Eq. (7), it is $\mathbf{P}_{ij}^{(SM)}$.
> - We thank the reviewer for pointing out the bald notation inconsistency issue with $P_{ij}^{\text{(SM)}}$. We fixed this issue in the revised version of the paper.
>
> > I am not sure where the proof of Theorem 3.4 is in the Appendix. Is it Theorem A.3?
> - We want to apologize regarding the inconsistent link to the proof of Theorem 3.4. Indeed, the proof is in the appendix A.3. We fixed this link in the revised version of the paper.
>
> [1] [Softmax action selection](https://en.wikipedia.org/wiki/Softmax_function#Reinforcement_learning)
>
> [2] [Value-Difference based Exploration](http://www.tokic.com/www/tokicm/publikationen/papers/KI2011.pdf)

---

### Author Response · Authors · 2023-11-20
**General response**

We appreciate all the reviewers for their time and insightful comments. We have responded to the individual reviews below and tried to address all reviewers' concerns. Here, we briefly summarize the changes we have made to the revised paper:

- We have added intuition explaining the exploration and exploitation of the self-attention in section 3.
- We have fixed bald notation in eq. 7.
- We have fixed the link to the proof of Theorem 3.4.
- We have added citation of the KVT Attention work in the related work section.
- We have improved Table 1 to clarify the different baselines of the Nystromformer method.

Additional experiments:

- We have added empirical justification for Fenton approximation and (Romeo) in Figure 6 of the Appendix.
The experiments section of Appendix 8 was extended with a stability plot showing the loss scale in Figure 8(b).
- We have added an evaluation of the Image Classification task in the experiments section in Appendix A.8.2.
- We have added results using the Long Range Arena benchmark in the experiments section in Appendix A.8.3

Additional ablation study:

- We conducted an ablation study to demonstrate the effect of temperature of LLN Attention on the accuracy and stability of the training. This new ablation study was added to Appendix A.9.4.

---

### Meta-Review · Area_Chair_9GmH · 2023-12-12

**Metareview:**

This paper introduces the Linear Log-Normal Attention (LLN Attention), a new approach aimed at addressing the quadratic time and memory complexity issues associated with self-attention in transformer architectures. The reviewers acknowledged that the theoretical analysis is thorough and empirical validation across various NLP tasks is extensive, demonstrating comparable performance to traditional softmax attention while benefiting from reduced computational complexity. However, there are also some concerns on that the intuitive explanations are lacking for some notions and the some NLP tasks are still missing (such as language modeling). Overall, this is a borderline paper, but may be useful for providing more discussions in the community.

**Justification For Why Not Higher Score:**

for some limitations, such as no language modeling evaluation.

**Justification For Why Not Lower Score:**

The topic is interesting and the paper has some merits.

---

### Decision · Program_Chairs · 2024-01-16

Accept (poster)